

# A global climatology of polar lows investigated for local differences and wind-shear environments

Patrick Johannes Stoll[1]

[1]Department of Physics and Technology, Arctic University of Norway, Tromsø, Norway

**Correspondence:** Patrick Johannes Stoll (patrick.stoll@uit.no)

**Abstract.**

Polar lows are intense mesoscale cyclones developing in marine polar air masses. This study presents a new global climatology of polar lows based on the ERA-5 reanalysis for the years 1979 - 2020. Criteria for the detection of polar lows are derived based on a comparison of six polar-low archives with cyclones derived by a mesoscale tracking algorithm. The characteristics associated with polar lows are considered by the criteria: (i) intense cyclone: large relative vorticity, (ii) mesoscale: small vortex diameter, and (iii) development in the marine polar air masses: combination of low dry-static stability and low potential temperature at the tropopause.

Polar lows develop in all marine areas adjacent to sea ice or cold landmasses, mainly in the winter half-year. The length and intensity of the season are regionally dependent. The highest density appears in the Nordic Seas. For all ocean sub-basins, forward-shear polar lows are the most common, whereas weak shear and those propagating towards warmer environments are second and third most frequent, depending on the area. Reverse-shear polar lows and those propagating towards colder environments are rather seldom, especially in the Southern Ocean. Generally, PLs share many characteristics across ocean basins and wind-shear categories. The most remarkable difference is that forward-shear polar lows are often occurring in stronger vertical wind shear, whereas reverse-shear polar lows feature lower static stability. Hence, the contribution to a fast baroclinic growth rate is slightly different for the shear categories.

## 1 Introduction

Polar lows (PLs) are intense mesoscale cyclones with a typical diameter of 200 - 500 km that develop when polar air masses advect over open water during the winter season (Rasmussen and Turner, 2003; Renfrew, 2015; Terpstra and Watanabe, 2020). PLs are one of the major natural hazards in the polar region, due to their gale-force winds (Wilhelmsen, 1985), large amount of snowfall (Harrold and Browning, 1969), low visibility, high waves (Orimolade et al., 2016), and potential for causing icing on ships and airplanes (Samuelsen et al., 2015).

To be alert of these destructive weather events, it is important to identify where and when PLs form. In this context, Stoll et al. (2018) has developed the first global climatology of PLs from the ERA-I reanalysis (Dee et al., 2011). The here-derived climatology features an update of the previous work by the following: The present effort (i) is based on the recently released ERA-5 reanalysis (Hersbach and Dee, 2016) at a horizontal grid spacing equivalent to 30 km as compared to 80 km of its





predecessor ERA-I, (ii) utilizes a tracking algorithm specifically tuned for the detection of mesoscale cyclones (Watanabe et al., 2016), and (iii) derives PL criteria from the comparison of six manually collected PL archives (Noer et al., 2011; Smirnova et al., 2015; Golubkin et al., 2021; Rojo et al., 2019; Verezemskaya et al., 2017; Yanase et al., 2016), instead of relying on a single such.

An atmospheric dataset that captures PLs is required for the derivation of a global PL climatology. Only recently atmospheric reanalyses have become sophisticated enough to include the majority of PLs. In the third version of the ECMWF reanalysis, ERA-40, Laffineur et al. (2014) identified 6 out of 29 PLs, whereas they detected already 13 of these PLs from the fourth version of the ECMWF reanalysis, ERA-Interim. More studies have estimated the fraction of represented PLs by ERA-Interim to 48% (Smirnova and Golubkin, 2017), 55% (Zappa et al., 2014), 60% (Michel et al., 2018) and 69% (Stoll et al., 2018).

These studies applied different detection methods for the PLs within ERA-Interim and utilized different amounts of cases. Impressively, the latest version of the ECMWF reanalysis, ERA-5, was found to reproduce most PLs, as it includes 93% of the PL events and 83% of the PL centers (Stoll et al., 2021). Also, the Arctic System Reanalysis (Bromwich et al., 2016) with a comparable horizontal resolution as ERA-5 reproduce most PL cases (Smirnova and Golubkin (2017): 89%, Stoll et al. (2018): 75%). In addition, ERA-5 provides the advantage of storing the output hourly, instead of 6-hourly for ERA-I, which improves

the tracking of fast-developing cyclones such as PLs.

The derivation of a PL climatology requires a methodology to identify PLs. However, the scientific community has so far not come to agreement on criteria for the definition of PLs (Moreno-Ibáñez et al., 2021). The most-accepted PL definition, formulated by Rasmussen and Turner (2003), is intensionally, due to the large variety of PLs, rather unspecific: "A polar low is a small, but fairly intense maritime cyclone that forms poleward of the main baroclinic zone (the polar front or other major

baroclinic zone). The horizontal scale of the polar low is approximately between 200 and 1000 km and has surface winds near or above gale force ($15\,\mathrm{m\,s^{-1}}$)."

This definition and recent encyclopedia entries on PLs (Renfrew, 2015; Terpstra and Watanabe, 2020) agree generally on the following PL characteristics: (i) being intense cyclone, (ii) the development in marine polar air masses poleward of the main baroclinic zone, and (iii) the mesoscale size. Criteria for these characteristics are to some degree arbitrary due to a smooth

transition between PLs and other cyclones. A subjective grey zone exists for PLs concerning their possible size, intensity, and air mass of development. However, a categorization into different types of cyclones is insightful for understanding their typical behavior and physical development. Such a categorization requires criteria, which may not fully exist in nature. Despite these challenges, this study provides a set of criteria successful to detect PLs.

A global climatology includes many PLs derived objectively and consistently. This allows for comparisons of PL activity

in different ocean sub-basins. Previous climatologies based on reanalyses (Zahn and von Storch, 2008a; Stoll et al., 2018) suggested that the Irminger Sea is the region with the highest PL density, but this was recently doubted by Golubkin et al. (2021) from manual identification of PL cases in the North Atlantic. PL climatologies enable the analysis of trends within the recent period of global warming. Zahn and von Storch (2008a) observed constant PL activity for the North Atlantic, Chen and von Storch (2013) detected a slightly increase of PLs in the North Pacific, and Stoll et al. (2018) found globally small changes

in PL frequency, but a decrease in the most intense cases. Further, a PL climatology eases the investigation of characteristic





environments and development mechanisms associated with PLs. For example, Chang et al. (2021) utilized the large amount of cases provided by a climatology to investigate how PLs are effected by of sudden stratospheric warmings, an event lasting only for a small fraction of the winter time.

PLs are developing in various vertical wind-shear environments (Duncan, 1978; Terpstra et al., 2016). The shear categorization into one weak-shear category and four strong-shear categories, being forward, left, reverse, and right, expresses large parts of the spatial variability in the PL environment and captures well the dynamics of the system (Stoll et al., 2021). However, the relative importance of the different shear categories has only been investigated for the Nordic Seas (Michel et al., 2018). Hence, this study investigates occurrences of the shear situations in the ocean sub-basins and for environmental differences in the shear categories.

The main research aims of this study are summarized as follows:

1. To derive a set of criteria that characterize polar lows.

2. To provide a climatology of polar lows to the scientific community.

3. To investigate differences among polar lows in the ocean sub-basins and among the vertical wind-shear categories.

Each of these aims is dedicated to one section (3 - 5). However, first the utilized data and methods are presented.

## 2    Data and methods

The approach for the derivation of the PL climatology is similar to Bracegirdle and Gray (2008) and Stoll et al. (2018). Cyclones are obtained by a tracking algorithm. The cyclones are compared to manually-detected PLs to obtain criteria that distinguish between PLs and other cyclones. The cyclones that satisfy the resulting criteria are forming the PL climatology.

### 2.1    Tracking algorithm

This study uses horizontal fields from the European Centre for Medium-Range Weather Forecasts (ECMWF) 5th reanalysis (ERA-5 Hersbach and Dee, 2016) for the years 1979 to 2020. ERA-5 provides hourly fields at a spectral truncation of T639, which is equivalent to a grid spacing of 30 km.

The PL climatology is dependent on the applied cyclone tracking algorithm. A study comparing the quality of different tracking approaches for capturing PLs has not been performed yet (Moreno-Ibáñez et al., 2021). Of primary importance for the tracking is the atmospheric variable used for the detection of the vortices. Multiple variables have been applied: the bandpass-filtered sea-level pressure (Zahn and von Storch, 2008b), the bandpass-filtered relative vorticity (Zappa et al., 2014; Yanase et al., 2016; Stoll et al., 2018), the smoothed Laplacian on the sea-level pressure (Michel et al., 2018), and the smoothed relative vorticity (Watanabe et al., 2016). The utilized variables are similar in attempting to measure the circulation of the flow and to emphasize the meso-scale component. Bandpass-filtered approaches may struggle to exclude synoptic-scale systems since higher harmonics are of considerable strength. Sea-level pressure and relative vorticity-based approaches are different



as the former detects the cyclone imprint at the surface, whereas the latter can be applied to the atmospheric level of interest. The sea-level pressure field is quite variable across PLs of different wind-shear categories (Stoll et al., 2021). In contrast, the mid-level relative vorticity and geopotential height anomaly field express considerably less spatial variability across PLs and therefore less prone to induce biases in the cyclone detection.

Hence, the mesoscale tracking algorithm developed by Watanabe et al. (2016), and adapted as described in Stoll et al. (2021), is applied to fields at a horizontal grid spacing of $0.25° \times 0.25°$. The algorithm detects local maxima exceeding $15 \times 10^{-5} \mathrm{s}^{-1}$ in the smoothed relative vorticity at 850 hPa, $\xi_{smth,850hPa}$. In this study, the relative vorticity is smoothed by a uniform filter of 60 km radius, since it improves the tracking result (Stoll et al., 2021) and produces results that are independent of the model resolution if the same filter is applied. At the consecutive time step, the tracking algorithm merges the largest vorticity maxima

occurring over open water within 150 km[1] of an estimated propagation by the mean wind of the 700 and 1000 hPa levels within 200 km distance. The built-in method of the algorithm by Watanabe et al. (2016) to exclude synoptic-scale disturbances is not used, since it excludes multiple PLs. Instead, the synoptic-scale disturbances are excluded later by the polar-low criteria.

Since PLs develop in marine polar air masses, they are expected neither over sub-tropical nor over ice-covered areas nor over landmasses. Hence, tracks are derived over open water in the latitude band $30° - 80°$ of both hemispheres. To avoid the

distortion of tracks by islands and peninsulas with a size of a few grid cells, these were defined as open water for the application of the tracking algorithm.

Only tracks with a lifetime of at least 6 h are kept. In total 300,000 cyclone tracks with 5.5 million time steps are obtained for the Northern Hemisphere, and 420,000 tracks with 8.4 million time steps for the Southern Hemisphere.

After investigation of the tracks, some issues are identified and solved by the following post-processing: Some tracks stay

along the domain boundary for multiple, consecutive time steps at the beginning or end of their lifetime. This occurs when the vortex maximum of the track is located outside, but near the domain boundary, such that the local maximum is identified at the boundary. Therefore, track segments are removed if they repeat location for at least 4 consecutive time steps, or are along the domain boundary for at least 6 consecutive time steps at the beginning or end of the lifetime. This excludes approximately 7.7% and 2.3% of the time steps in the Northern and Southern Hemisphere, respectively.

Multiple tracks stay in the vicinity of land for most of the lifetime, due to: (i) vorticity anomalies induced by orography, or (ii) vortex centers located over land, which are identified at the coast, since the land is masked. Most of these tracks are excluded by the following criteria: The track must remain within two grid cells from land for more than 50% of their lifetime. This removes 18.5% of time steps for the Northern and only 2.7% for the Southern Hemisphere, since fewer land interfere with the tracks in the latter hemisphere.

Some tracks were found to re-intensify considerably after having decayed. Tracks are divided if they have two local maxima in $\xi_{smth,850hPa}$ exceeding $25 \times 10^{-5} \mathrm{s}^{-1}$ and the local minima in between is at least 40% lower than the weaker of the two maxima. This is only applied to approximately 0.4% and 0.3% of the tracks in the Northern and Southern Hemisphere, respectively, but for some of the matched PL tracks presented in the next section.

---

[1]Stoll et al. (2021) merged the closest vorticity maxima within 100 km, but some testing found the applied values slightly superior for this study.



**Table 1.** Match statistics of the different PL lists to ERA-5 cyclone tracks. The second column provides the number of PLs in each list, the third column the number of PL tracks with more than three time steps, considered necessary for a trustworthy matching. The Yanase is treated differently since it provides only one track point per PL. The number and fraction of PLs matched by a ERA-5 cyclone track is presented in column 4 and 5, respectively. A match is obtained if the cyclone track is within a distance of 150 km of the PL for at least half of the track points of the PL. Column 6 displays the amount of matched cases that are excluded since they start or end more than 24 h earlier or later than the PL from the list, and the number of ERA-5 tracks match to two PLs from the list. The last column provides the amount of matched PLs included in the parameter derivation.

| Polar-low list | number cases | cases > 3 steps | ERA-5 matched | % | excl. >24h | excl. 2x | remaining matches |
|---|---|---|---|---|---|---|---|
| Noer | 185 | 162 | 131 | 81 | 32 | 1 | 98 |
| Rojo | 420 | 391 | 255 | 65 | 38 | 4 | 213 |
| Smirnova | 637 | 251 | 153 | 61 | 54 | 0 | 99 |
| Golubkin | 131 | 123 | 80 | 65 | 15 | 0 | 65 |
| Yanase | 19 | (none) | 13 | 68 | 0 | 0 | 13 |
| Verezemskaya | 1735 | 1139 | 363 | 32 | 93 | 13 | 257 |

After the post-processing, approximately 210,000 cyclone tracks with 4.1 million time steps remain for the Northern and
390,000 cyclones tracks with 8.0 million time steps for the Southern Hemisphere for 42 years of reanalysis data.

## 2.2 Subjective polar-low lists

A novelty of this study is the comparison of five lists of manually-detected PLs and one of mesocyclones to derive characteristic criteria for PLs. To the best knowledge of the author, these are all available lists of such systems that include spatio-temporal information sufficient for tracking. In the following, the lists are called after the first author of the presenting scientific study.

The **Noer list** represents the 2011 version of the STARS dataset[2]. The STARS dataset provides the primary PL center of cases operationally collected by the Norwegian Meteorological Institute based on inspection of satellite imagery and investigation of the synoptic-scale conditions by their current weather prediction model (Noer et al., 2011). Track points are at hourly resolution due to interpolation between locations identified from satellite images. The Noer list contains 185 cases during the years 2002 - 2011 from the North-Eastern Atlantic. It has been extensively utilized by multiple studies investigating PLs (e.g. Zappa et al.,
2014; Laffineur et al., 2014; Terpstra et al., 2016; Michel et al., 2018; Stoll et al., 2018).

The **Rojo list** is a recent update of the STARS dataset for the years 1999 - 2019 (Rojo et al., 2015, 2019; Rojo et al., 2019). Additionally to the primary centers originally listed in the STARS dataset, the Rojo list includes individual centers for situations of multiple PLs, such that it consist of 420 PL centers, mainly from the Nordic Seas, but with a few cases to the west of Iceland

---

[2]https://projects.met.no/polarlow/stars-dat/





and the British Isles. Comparison to the Noer list for the overlapping period reveals that some tracks are considerably different, such that the inclusion of both lists appears reasonable even though they are not completely independent of each other.

The **Smirnova list** provides PL cases for the Nordic Seas mainly North of 70° N of the years 1995 - 2009 obtained from a combination of different satellite products (Smirnova et al., 2015). It contains in total 637 PLs, hence more cases per season for a smaller area than the Rojo list, which may be due to the inclusion of weaker systems. Accordingly, only 39% of the PLs from this list persist of more than three time steps (Table 1), whereas around 90% of the PLs from the other lists exceed three track points.

The **Golubkin list** contains 131 PLs of the year 2015 - 2017 for the whole North Atlantic, including the Labrador, Irminger, and Nordic Seas (Golubkin et al., 2021). It was derived by inspection of different satellite imagery combined with synoptic weather charts. All of the previous three lists contain track time steps when the PL was identified on a satellite image, hence at irregular intervals up to 12 h.

The **Yanase list** is a collection of 19 PLs investigated in literature in the past four decades in the Sea of Japan (Yanase et al., 2016). Different from all other lists, it contains only one time step with a location for each PL, such that matching as presented in the next section is challenging. However, this list represents PLs from a different ocean basin than the North Atlantic.

The **Verezemskaya list** collects 1735 mesoscale cyclones in the Southern Ocean in June - September 2004 (Verezemskaya et al., 2017). The mesoscale cyclones are detected from satellite infra-red imagery. However, the list neither provides information on whether a system is intense enough for being considered PLs, nor does it ensure the development in the polar air masses. Hence, this list is just utilized for a broad comparison to the PL lists from the Northern Hemisphere.

## 2.3 Track matching

In order to investigate the characteristics of PLs, which are utilized as PL criteria, the ERA-5 representations of the tracks from the PL lists are identified. Only tracks from the PL lists with more than three time steps are considered, to ensure a trustworthy track matching. An exception is the Yanase list since it only provides one time step for each PL.

The following definition for a match is applied: A cyclone track matches to a track from a PL list, if the cyclone track is located within 150 km for at least half of the track points of the PL. This ensures a rather strict spatio-temporal agreement of the tracks but is still flexible for some inaccuracies (i) in the reanalysis to reproduce the PL at the correct location, (ii) in the cyclone tracking algorithm to detect the PL, and (iii) in the location of the PL in the manually-derived lists. Different merging distances were tested (Table S 1). A distance of 100 km significantly reduces the detection rate as compared to 150 km distance. A distance of 250 km results in slightly higher detection rates, but at a lower quality of the matches.

For all PL lists of the Northern Hemisphere the match rate to ERA-5-based cyclone tracks is quite high (Tab. 1, Col. 5). More than 80% of the PLs from the Noer list are matched and between 60 and 70% for the Rojo, Smirnova, Golubkin and Yanase lists. The Noer list likely has the highest match rate since it includes major PL centers, that are detected routinely during operational weather prediction, whereas the other lists either include secondary PL centers (Rojo list) or solely rely on satellite images (e.g. Smirnova list), which both may result in the inclusion of weak systems and problems in the tracking due to large time gaps between images.



For the mesocyclone list of Verezemskaya from the Southern Ocean, only 32% of the tracks match a cyclone track. This can be explained by the list consist of mesocyclones, which are by definition not required to be intense. The high rate of weak systems in the Verezemskaya list is also expressed by a large number of cases in a short time period of four months as compared to the other PL lists that span multiple years. Hence, most systems in the Verezemskaya list do not exceed the vorticity threshold used in the tracking algorithm, which is tuned for detecting cyclones with an intensity expected for PLs.

The matched tracks to the PL lists are utilized for the derivation of identification criteria for PLs. However, some matched tracks have a considerably longer lifetime than the corresponding track from the PL list (Tab. 1, Col. 6). These tracks feature PLs intensifying from a pre-existing circulation anomaly, such as a frontal zone or a renascent of an extra-tropical cyclone. Such transitions are considered part of the life cycle of some PLs, hence are not necessarily false positives. However, systems that are non-PLs for part of their lifetime may distort the derivation of the PL parameters. Therefore, PL-matched cyclone tracks that begin or end 24 h earlier or later, respectively, than the PL from the list are excluded for the parameter derivation. This excludes approximately one-quarter of the matched tracks from the different lists.

A few tracks match two PL tracks, due to multiple PLs with several centers in close vicinity. In such cases, only one matched track is kept, which excludes a few tracks (Tab. 1, Col. 7). Manual inspection of the remaining matched PL tracks reveals that they are good representations of the PLs from the lists. For simplicity, the matched PL tracks are in the following only referred to as PLs or PL tracks.

## 2.4 Parameter derivation

Parameters in the vicinity of the cyclone and PL tracks are investigated. Most parameters are computed as the local mean within 250 km to the center. To save disk space the parameters are derived from ERA-5 horizontal fields at a $0.5° \times 0.5°$ grid spacing. The reduced grid spacing was found to produce similar results for the computation of the local averages. Differently, the near-surface wind speed, $U_{10m}$, which is computed as the local maxima within 250 km to the cyclone center, is derived from ERA-5 fields at a $0.25° \times 0.25°$ grid spacing, to prevent smoothing of the local wind maxima.

The tropopause is defined as the lowest atmospheric level where 2 PVU (1 PVU = $10^{-6}$ K m$^2$ kg$^{-1}$ s$^{-1}$) is reached. The maximum tropopause wind speed poleward of the system is derived as the maximum value within a band $\pm 0.5°$ longitude and at a latitude with at least the same magnitude as the location of the track.

The parameters are computed for each time step of the track. For the comparison of PL-matched tracks and all cyclone tracks, the lifetime-mean, maximum, and minimum are compared and the most successful is chosen.

The **vertical wind shear** is derived as in Stoll et al. (2021), but within a distance of 250 km, instead of 500 km. Both radii were tested and the results are robust, but the smaller radius appears to better represent the local environment. The vertical wind-shear angle is computed between the tropospheric mean wind vector and the wind-shear vector, as in Terpstra et al. (2016). This is different from Stoll et al. (2021) who analyze PLs from the perspective of their propagation direction and hence compare the shear vector to the propagation of the PLs. However, Stoll et al. (2021) argue that the mean tropospheric wind provides a good estimate of the propagation direction of PLs, and hence both methods are expected to give similar results. Differences in the propagation direction and the orientation of the mean tropospheric wind vector can occur in situations of





slow propagation associated with low wind speeds. Different to Stoll et al. (2021), the upper threshold for the weak shear category is set to 1.0 instead of $1.5 \times 10^{-3} \mathrm{s}^{-1}$, since the purposes are different. Stoll et al. (2021) demonstrated differences among the strong shear categories, whereas this study aims to identify the locations in which the shear types occur. However, the results of both studies are robust for the other threshold.

The **vortex diameter** is estimated by computing the diameter of a circle with the same area as the vortex area, which is the area surrounding the vorticity maxima with $\xi_{smth,850hPa}$ exceeding $10 \times 10^{-5} \mathrm{s}^{-1}$ as described in Watanabe et al. (2016) but with thresholds in accordance to Stoll et al. (2021). It should be noted that the vortex diameter depends to some degree on the threshold and the smoothness of the vorticity field.

## 3 Polar-low criteria

Different parameters are compared for their ability to separate between PLs and other cyclones. The following set of criteria is successful to detect PLs, and is in the following called the PL criteria:

1. Polar-front criterion: $\theta_{trop} < 300.8 \, \mathrm{K}$

2. Static-stability criterion: $\theta_{500hPa} - \theta_{SST} < 11.0 \, \mathrm{K}$

3. Intensity criterion: $\xi_{smth,850hPa} > 20.0 \times 10^{-5} \mathrm{s}^{-1}$

4. Mesoscale-size criterion: Vortex diameter $< 430 \, \mathrm{km}$.

A track that satisfies the first in the mean of the lifetime, the second as the minimum and the third and forth as maximum is defined a PL track. As noted earlier, PLs are sometimes transitioning from or towards other types of cyclones, hence they are not necessarily PLs for their entire lifetime. Therefore, PL time steps are introduced as the parts of the PL track that satisfy all four criteria simultaneously. 51% of the time steps from the PL tracks in the Northern and 37% in the Southern Hemisphere are PL time steps. Most PL tracks satisfy all criteria simultaneously for parts of their lifetime.

Following consideration is used for the derivation of the PL criteria: Only a small fraction of all cyclone tracks is expected to be PLs, hence, a criterion is considered successful if it holds for most PLs but only for a few cyclones. This set of criteria retains most PLs from the lists (68 - 89%), whereas only a small fraction of cyclones remains, being 6.7% for the Northern and 3.6% for the Southern Hemisphere (Tab. 2, all years). These remaining tracks form the PL climatology presented in the next section. However, first the derivation of the PL criteria is presented.

The threshold for a criterion (Tab. 3 Col. 2) is defined such that it excludes less than 10% of the PLs of all lists (Col. 3), with a small exception for the static-stability criterion explained in Section 3.1.2. The Verezemskaya list of mesocyclones does not contribute to the parameter derivation since it contains too many non-PLs. However, it is also discussed whether the mesocyclones share qualitatively similar characteristics to the PLs. The successfulness of a criterion is measured by the fraction of all cyclones it excludes (Col. 4). The additional value of a criterion is examined by the excluded fraction of cyclones only excluded by this criterion, but not by the three criteria of other types (Col. 5).





**Table 2.** Statistics for the satisfaction of the polar-low criteria from the different polar low lists and all cyclone tracks of the Northern (NH) and Southern Hemisphere (SH).

| Polar-low list | Number of tracks | Satisfying all criteria | % |
|---|---|---|---|
| Noer | 98 | 80 | 82 |
| Rojo | 213 | 184 | 86 |
| Smirnova | 99 | 68 | 69 |
| Golubkin | 65 | 58 | 89 |
| Yanase | 13 | 9 | 69 |
| Verezemskaya | 257 | 50 | 20 |
| Cyclone tracks | | | |
| NH (2009) | 4,905 | 377 | 7.7 |
| SH (2004) | 9,530 | 321 | 3.4 |
| NH (1979 - 2020) | 207,748 | 13,888 | 6.7 |
| SH (1979 - 2020) | 391,801 | 14,041 | 3.6 |

The parameters for cyclone tracks are derived for one year of each hemisphere only, to save computational resources. Tests show that the cyclone statistics are robust between years, hence this does not influence the results. For the Northern Hemisphere, 2009 is randomly chosen, and for the Southern Hemisphere, 2004 is taken since it is the same year as the list of Verezemskaya, allowing for the double usage of reanalysis data required for the parameter computation.

Further, it is ensured that the criteria consider all the characteristics assigned to PLs by the scientific community, which are: intense, mesoscale, and development in marine polar air masses. Another aim is that the PL criteria are universally applicable independent of the region.

## 3.1 Marine polar-air masses

A guideline for the identification of the marine polar-air masses poleward of the main baroclinic zone is not provided by any PL definition. This study exploits two characteristics of the marine polar-air mass for its identification: (i) location poleward of the polar front, leading to criterion 1, and (ii) a low dry-static stability, resulting in criterion 2. The two criteria favor each other, which is expressed by a correlation in the parameters for the cyclone tracks in the Northern Hemisphere of 0.73. However, Stoll et al. (2018) demonstrate that approximately one-third of the cyclones with low static stability still feature a considerable jet on the poleward side, therefore utilizing two criteria for the detection of the marine polar-air masses. Also, this study finds that two criteria in collaboration are more successful in discriminating between PLs and other cyclones than is a single criterion (see Fig. S 1). Further, criteria 1 and 2 are different. Criterion 1 is computed as the lifetime-mean, hence it ensures that the PL is





**Figure 1.** Distribution in the parameters used to identify polar lows, these are the polar-low criteria which are successful in distinguishing between all tracks of both hemispheres (black) and polar low from the different lists (colours). The tracks that remain after application of the other three polar-low criteria are presented (black dashed-dotted and dotted) to express the additional value of the parameter to the other three criteria. (a) The potential temperature at the tropopause, $\theta_{trop}$, is computed as lifetime-mean of the track, (b) the potential static stability, $\theta_{500hPa} - \theta_{SST}$, as lifetime-minimum. (c) The relative vorticity and (d) the vortex diameter expresses the lifetime-maximum. The coloured triangle along the x-axis denotes the threshold satisfied by 90% of the polar lows of each list.

in the polar-air mass for most of its lifetime. In contrast, criterion 2 is derived as the lifetime-minimum and ensures low static
stability at least once during the PL development, likely in the intensification phase.



**Table 3.** Statistics for the derivation of the polar-low criteria. The type expresses whether the parameter is computed as lifetime-mean, maximum or minimum of a track. The threshold is chosen such that it is satisfied by 90% of the polar lows of all lists, not including the Verezemskaya list of mesocylones. For $\theta_{500hPa} - \theta_{SST}$ in addition also the threshold satisfying all polar-low lists beside the one from Smirnova is presented. The thresholds displayed in red mark the polar-low criteria. Column 4 expresses the fraction of polar lows from the six lists excluded by the threshold. Column 5 provides the fraction of cyclones excluded by the threshold. The last column displays the additionally excluded cyclones by the threshold after application of the polar-low criteria of different type. The bolt headlines separate the types of criteria.

| Parameter | Type | Threshold | excluded polar lows [%] N/R/S/Y/G/V | excluded cyclones NH/SH [%] | excl. cycl. after crit. NH/SH [%] |
|---|---|---|---|---|---|
| **Polar-front criterion** | | | | | |
| $\theta_{trop}$ | mean | $< 300.8\,\text{K}$ | 9/4/5/8/2/40 | 76/ 65 | 14/ 14 |
| $U_{trop,polew}$ | mean | $< 41.2\,\text{m s}^{-1}$ | 4/1/3/8/3/35 | 51/ 47 | 16/ 18 |
| **Static-stability criterion** | | | | | |
| $SST - T_{500hPa}$ | max | $> 38.4\,\text{K}$ | 0/0/9/8/2/43 | 69/ 68 | 17/ 46 |
| $\theta_{500hPa} - \theta_{SST}$ | min | $< 13.2\,\text{K}$ | 3/1/9/8/2/41 | 75/ 69 | 18/ 33 |
| | min | $< 11.0\,\text{K}$ | 7/4/17/8/3/56 | 80/ 74 | 34/ 55 |
| $\theta_{500hPa} - \theta_{925hPa}$ | min | $< 20.3\,\text{K}$ | 1/1/3/8/0/5 | 49/ 36 | 4/ 3 |
| **Intensity criterion** | | | | | |
| $\xi_{smth,850}$ | max | $> 20 \times 10^{-5}\text{s}^{-1}$ | 0/1/9/8/5/5 | 19/ 19 | 20/ 20 |
| $U_{10m}$ | max | $> 14.1\text{m s}^{-1}$ | 4/3/7/8/3/3 | 21/ 12 | 13/ 8 |
| **Meso-scale size criterion** | | | | | |
| Vortex diameter | max | $< 430\,\text{km}$ | 9/8/9/8/5/44 | 25/ 34 | 24/ 40 |
| **Other parameters** | | | | | |
| Lifetime | | $> 12\,\text{h}$ | 9/7/6/8/8/7 | 39/ 38 | 23/ 17 |
| Distance to land | max | $> 140\,\text{km}$ | 10/7/7/46/9/0 | 15/ 3 | 21/ 2 |
| Sea-level pressure | min | $< 1010.5\,\text{hPa}$ | 2/3/2/8/5/0 | 13/ 6 | 3/ 1 |

### 3.1.1 Polar-front criterion

PLs are defined to develop poleward of the polar front (Rasmussen and Turner, 2003). The potential temperature at the tropopause, $\theta_{trop}$ can be used for separation between polar and more temperate air masses. A low potential temperature at the tropopause is reached by a combination of a low temperature at the tropopause and a low altitude of the tropopause, both favored in polar-air masses.

All PL lists agree that the lifetime-mean in $\theta_{trop}$ is considerably lower for PLs than for most cyclones (Fig. 1a). A threshold that excludes less than 10% of PLs for all lists is found at 300.8 K (Tab. 3). Also, the mesocyclones from the Southern Hemi-





sphere (Verezemskaya) have a lower potential temperature at the tropopause than general cyclones but have higher values than the Northern Hemisphere PLs.

The polar-front criterion, $\theta_{trop} < 300.8$ K, individually excludes 76% and 65% of the Northern and Southern Hemisphere cyclone tracks, respectively, expressing its high value. As mentioned before, this criterion is dependent on the static-stability criterion, but the polar-front criterion excludes additional 14% of tracks for both hemispheres (Tab. 3, Fig. 1a, dashed and dotted).

    Stoll et al. (2018) use the maximum tropopause wind speed poleward of the system, $U_{trop,polew.}$, to identify systems pole-
ward of the polar front. The lifetime-mean in $U_{trop,polew}$ is considerably lower for PLs, mainly below $40 \, \mathrm{m \, s^{-1}}$, than for cyclone tracks, where it ranges from 20 to $80 \, \mathrm{m \, s^{-1}}$ (Fig. 2a). Hence, $U_{trop,polew.}$ has a potential to separate between PLs and other cyclones. $U_{trop,polew}$ is less successful as individual criterion than $\theta_{trop}$ (Tab. 3, Col. 5), but slightly superior over $\theta_{trop}$ after application of the other three criteria (Col. 6). However, distributions in $U_{trop,polew}$ are considerable different across PL lists (Fig. 2a), indicating a local dependence of this criterion, which is less the case for $\theta_{trop}$. For the PLs from the Sea of
Japan (Yanase list), the threshold is at $41 \, \mathrm{m \, s^{-1}}$, whereas for lists of PLs in the Nordic Seas (Rojo, Smirnova) the threshold is at $29 \, \mathrm{m \, s^{-1}}$, close to the one used in Stoll et al. (2018). The higher threshold in $U_{trop,polew}$ for the Sea of Japan is likely due to some PLs at the lower latitude of the Sea of Japan occurring in polar air masses characterized with a meandering jet stream such that the jet is located poleward of the system when measured along the same longitude as done for the computation of $U_{trop,polew}$. Therefore, $\theta_{trop}$ is chosen as criterion, since it is shows smaller regional dependency than does $U_{trop,polew}$. The
distributions in $U_{trop,polew}$ for the remaining cyclone tracks after application of the four PL criteria are in good agreement with the distributions of the PLs (Fig. 2a). This expresses that the polar-front characteristics are captured by the PL criteria.

### 3.1.2  Static-stability criterion

Multiple studies have found that PLs form in environments of low dry-static stability through considerable depth of the troposphere (Forbes and Lottes, 1985; Noer et al., 2011; Stoll et al., 2018; Terpstra et al., 2021). This is applied for detection of
PLs in several studies as a large vertical temperature contrast in the PL environment between the sea surface and the 500 hPa level ($SST - T_{500hPa}$ Zahn and von Storch, 2008a; Chen et al., 2014; Zappa et al., 2014; Yanase et al., 2016). Stoll et al. (2018) found that the dry-static stability is superior to the moist-static stability in distinguishing between PLs and other cyclones, and that it is more successful when measured between the sea-level and the 500 hPa level than when 700 hPa or 850 hPa are used as the upper level. Low dry-static stability is characteristic for the marine polar air masses since marine air masses warmed
from the sea feature moist-adiabatic lapse rates, which converge towards the dry adiabats at low temperatures typical for the winter-time polar regions (Stoll et al., 2021).

    The investigated parameters measuring the static stability, $\theta_{500hPa} - \theta_{SST}$, $SST - T_{500hPa}$, and $\theta_{500hPa} - \theta_{925hPa}$ are all found to be successful in distinguishing between PLs and cyclones (Figs. 1 and 2). This confirms that PLs are developing in environments of low dry-static stability of considerable depth.

$\theta_{500hPa} - \theta_{SST}$ and $SST - T_{500hPa}$ are superior to $\theta_{500hPa} - \theta_{925hPa}$ in excluding cyclone tracks (Table 3). $\theta_{500hPa} - \theta_{925hPa}$ provides a direct measure of the static stability. In contrast, $\theta_{500hPa} - \theta_{SST}$ and $SST - T_{500hPa}$ indicate the "poten-





tial" static stability of the troposphere. "Potential" since the sea-surface temperature is considerably warmer than the low-level atmosphere by sometimes 10 K in strong marine cold-air outbreaks (e.g. Papritz et al., 2015). Hence, the two parameters utilizing the $SST$ do not measure the actual static stability of the troposphere, but the static stability if the lower troposphere would
be heated to the sea-surface temperature. This emphasizes that heating of the lower atmosphere from the sea is characteristic for PL environments.

The two measures for the potential static stability, $\theta_{500hPa} - \theta_{SST}$ and $SST - T_{500hPa}$, are quite similar in the success of excluding cyclone tracks. However, the following consideration leads to the choice for the former: $\theta_{500hPa} - \theta_{SST}$ corrects the potential static stability for pressure variations at sea level, which is ignored by $SST - T_{500hPa}$. High values in $SST - T_{500hPa}$
are supported by a high sea-level pressure, when the vertical distance between the sea surface and the 500 hPa level is large.

The primary characteristics of PLs appear to be the low potential dry-static stability, whereas the environmental-mean in the sea-level pressure, a synoptic-scale parameter, does not have a direct physical impact on PL development (Section 3.4). Monthly mean values in the sea-level pressure are varying between seasons and ocean basins. For example, the wintertime high latitudes are rather of low sea-level pressure, a reason why $\theta_{500hPa} - \theta_{SST}$ in isolation is slightly more successful for
detection of PLs than $SST - T_{500hPa}$. However, the polar-front criterion is mainly satisfied within the winter season, making for the Northern Hemisphere both criteria equally successful in excluding cyclones remaining after application of the other three PL criteria (Tab. 3 Col. 5). Differently, in the Southern Ocean, $SST - T_{500hPa}$ excludes considerably more remaining cyclones than $\theta_{500hPa} - \theta_{SST}$ (Col. 5), since it is the area of globally lowest sea-level pressure, contributing to lower values in $SST - T_{500hPa}$ and hereby "punishing" this region if a criterion with $SST - T_{500hPa}$ is used. Hence, it appears that
$\theta_{500hPa} - \theta_{SST}$ is less prone to induce seasonal and regional biases than $SST - T_{500hPa}$.

All PL lists of the Northern Hemisphere, beside the one from Smirnova agree on a stricter threshold $\theta_{500hPa} - \theta_{SST} < 11.0$ K instead of 13.2 K. Also, most PLs from the Smirnova list satisfy this stricter criterion (83%), however, the Smirnova list includes some cases with considerably higher static stability, which may be false positives. The stricter threshold in $\theta_{500hPa} - \theta_{SST}$ is considerably more effective in excluding additional cyclones than the weaker threshold of 13.2 K, and therefore chosen here.
The distribution of $\theta_{500hPa} - \theta_{925hPa}$ becomes similar for the cyclone tracks satisfying the PL criteria and the PLs from the lists (Fig. 2c), indicating that the PL criteria capture the static-stability characteristics of PLs in addition to the "potential" static stability.

The threshold of $\theta_{500hPa} - \theta_{SST} < 11.0$ K is approximately equivalent to $SST - T_{500hPa} > 40$ K for the Northern Hemisphere, but approximately at 38 K for PLs in the Southern Ocean which often features a low sea-level pressure (Fig. 2b). Hence,
the criterion on the static stability is lower than the threshold of $SST - T_{500hPa} > 43$ K utilized by multiple studies (Zahn and von Storch, 2008a; Zappa et al., 2014; Yanase et al., 2016). However, Terpstra et al. (2016) noted that a high static-stability threshold may bias a PL dataset towards reverse-shear cases since more forward-shear PLs are excluded. The weaker threshold applied here is less prone to biases in shear situations.





## 3.2 Intensity criterion

PLs are intense meso-scale cyclones. The definition by Rasmussen and Turner (2003) provides an intensity threshold by the near-surface wind speed, $U_{10m}$ exceeding $15 \, \mathrm{m \, s^{-1}}$ in the vicinity of the PL, which is commonly used for the detection of PLs (Zappa et al., 2014; Yanase et al., 2016; Verezemskaya et al., 2017, e.g.). However, Noer et al. (2011) note that environmental air masses around PLs frequently advect at a similar velocity. They instead consider the local wind enhancement to measure the intensity of a PL. Accordingly, Stoll et al. (2018) find that the local cyclone depth and the relative vorticity are superior

criteria for the PL detection as compared to that of the near-surface wind speed.

The cyclone tracks and the PLs have similar distributions in the near-surface wind speed (Fig. 2d), indicating that $U_{10m}$ is poor in distinguishing strong from weak mesoscale cyclones being embedded in a strong background flow.

A more successful parameter to measure the intensity of PLs is the smoothed relative vorticity at $850 \, \mathrm{hPa}$, $\xi_{smth,850hPa}$, which provides a measure of the local vortex strength independent of the background flow. $\xi_{smth,850}$ is larger for the PLs from

the lists than for cyclone tracks (Fig. 1). The PL lists agree on a threshold $\xi_{smth,850hPa} > 20.0 \times 10^{-5} \mathrm{s^{-1}}$, which excludes 20% of the cyclones for both hemispheres not excluded by the other PL criteria.

Distributions in $U_{10m}$ become similar for PLs and the tracks satisfying the PL criteria (Fig. 2d), and the threshold of $15 \, \mathrm{m \, s^{-1}}$ defined by Rasmussen and Turner (2003) is mainly satisfied for the tracks satisfying the criteria. Hence, $U_{10m} > 15 \, \mathrm{m \, s^{-1}}$ may be considered necessary due to the definition of Rasmussen and Turner (2003), but it is not a sufficient criterion for identifying

PLs and can be ignored when utilizing the relative vorticity.

## 3.3 Meso-scale size criterion

PLs are characterized by their meso-scale size, however, the transition from meso to synoptic-scale cyclones is seamless. The definition by Rasmussen and Turner (2003) specifies a spatial range between 200 and $1000 \, \mathrm{km}$. However, observational PL lists contain few systems with diameters larger than $600 \, \mathrm{km}$ (Rojo et al., 2015; Blechschmidt, 2008). A general method for

measuring the size of cyclones across the meso and synoptic-scale is not established. Observational lists are typically deriving the size from the PL-associated clouds (Rojo et al., 2015). For automation purposes, this approach is problematic when the cyclone intersects with adjacent clouds, often the case in PL environments. Closed pressure contours may be a reasonable measure for synoptic-scale cyclones (e.g. Simmonds and Keay, 2000), however, the pressure contours of meso-scale systems are often distorted by the environmental flow. To circumvent the influence of a uniform background flow, Watanabe et al.

(2016) introduce the vortex area defined by the adjacent region with high relative vorticity.

The vortex diameter is generally below $430 \, \mathrm{km}$ for the PLs from the different lists, whereas it is considerably larger for some cyclone tracks (Fig. 1d). A vortex diameter below $430 \, \mathrm{km}$ excludes an additional 24 and 40% of the cyclones of the Northern and Southern Hemisphere, respectively. Hence, it is a useful criterion for the identification of PLs.

The mesocyclones from the Verezemskaya list include multiple large systems, which appear not to be PLs. This provides

another argument for not using this list for the derivation of the PL parameters.





## 3.4 Comparison to other parameters

Parameters that are not contributing to the PL criteria are compared for the cyclone tracks and the PLs from the lists (Fig. 2). In general, the distributions in these parameters become quite similar for the PLs and the tracks which are satisfying the PL criteria, which gives confidence that the criteria are skillful. The first four parameters of Fig. 2 were discussed in the previous
sections, the remaining are presented in the following.

Surprisingly, the lifetime of cyclones is shorter than for the PLs from the lists (Fig. 2e), even though PLs are known for their rather short lifetime as compared to extra-tropical cyclones. This pinpoints that the tracking algorithm is successful in targeting mesoscale cyclones of a short lifetime. The lifetime of a track is correlated to the maximum intensity of the track (0.44), a criterion for PL identification. This indicates that the PLs are among the meso-scale cyclones of a longer lifetime.

The tracks satisfying the PL criteria have slightly shorter lifetime than the PLs from the lists. The small difference may be explained that PLs from the lists being biased towards longer lifetimes, since the PLs from the lists require a match in ERA-5, which is more likely the case for PLs with a longer lifetime, which are better simulated by ERA-5 and captured by the tracking and matching procedure.

The sea-level pressure of all cyclone tracks is slightly higher than for the PLs within each hemisphere. However, PLs from the
Yanase list have slightly higher sea-level pressure than the cyclones tracks of the Northern Hemisphere. Hence, the distribution in the sea-level pressure is dependent on the region, as noted in the argumentation for choosing $\theta_{500hPa} - \theta_{SST}$ instead of $SST - T_{500hPa}$.

Tracks satisfying the PL criteria have similar sea-level pressure distributions to the PLs of the same hemisphere. This indicates that the tracks captured by the PL criteria feature similar characteristics to the PLs, however, the sea-level pressure is
not of value for the detection of PLs.

## 3.5 Validation: Misses and False Positives

Since the scientific community does not agree on criteria for the detection of PLs (Moreno-Ibáñez et al., 2021), the estimation of miss and false-positive rates is subjective. Still, estimates in these rates are important for expressing the quality of the derived PL climatology. To the author's knowledge, it is the first time these rates are estimated for a PL dataset.

First, the degree of subjectivity is demonstrated by a comparison of the manually-derived PL lists for times and regions of spatio-temporal overlap. The miss and false-positive rates are estimated by defining "list a" as ground truth. Then cases from "list a" missing in "list b" are misses, and cases in "list b" not in "list a" false positives. Setting the Rojo list as ground-true, the Golubkin and Smirnova list have miss rates of 26% and 78%, and false-positive rates of 38% and 74%, respectively.

For the PL climatology, the misses are estimated by combining the match statistics between the PL lists and ERA-5 (Tab. 1)
with the fraction of systems satisfying the PL criteria (Tab. 2). For the different PL lists, 61% (Smirnova) to 81% (Noer) of the PLs were matched with a track in ERA-5, with the mean of the list being 68%. From the matched tracks 68% (Smirnova) to 89% (Golubkin) satisfy the derived PL criteria, with a mean of the lists being 79%. Hence, the derived climatology contains between 42% (Smirnova) and 66% (Noer) of the PLs from the lists, with a mean of the lists being 54%. Hence, the climatology





miss around 46% of the PLs from the PL lists. The miss rate appears rather high, however well within the miss rates when
different PL lists are compared to each other, as demonstrated above.

The largest contribution to the miss rate is due to the track matching. On average over the lists, 32% of the PLs do not have a match in ERA-5. Possible reasons are: (i) ERA-5 does not simulate the PL at the correct location at a sufficient strength, (ii) the tracking procedure is not capable to reproduce the track of every PL, (iii) uncertainty in the track location of the PLs from the lists, (iv) some degree of subjectivity whether all systems in the lists are clear-cut PLs. All these reasons appear to contribute,
whereas quantification of their relative importances is difficult. The applied method for the track matching is rather strict. If the match distance is relaxed from 150 to 250 km, the average match rate of the lists increases from 68% to 73%, such that the total miss rate decreases from 46% to 42%.

The other contribution to the miss rate is the application of the PL criteria, which as mean of the PL lists excludes 21% of the PLs. Possible reasons: (i) the criteria are too strict, (ii) some degree of subjectivity whether all systems in the lists are clear-cut
PLs. The method for the derivation of the PL criteria is a compromise between excluding some matched PLs versus excluding as many non-PL cyclone tracks as possible (Tab. 1); hence weaker criteria come at the expanse of more false positives.

For the evaluation of false positives, 100 of the detected PLs were randomly chosen and investigated whether they are considered PLs. For the years 1980 and 2013, 30 and 20 PLs were analyzed for the Northern and Southern Hemisphere, respectively, approximately representing the share of PL time steps from the two hemispheres.

The author considers around 80% of the investigated cases of both hemispheres as reasonable detections of PLs. Around 10% appear as false positives and another 10% as borderline cases. The false positives are extra-tropical cyclones, frontal zones, and orographically induced shear zones. The border-line cases consist of systems that satisfy the PL criteria only for a short part of their lifetime and originated from, or were propagating towards temperate air masses. Some cases are borderline since they are intense meso-scale cyclones, but close to the main baroclinic zone or to the center of an extra-tropical cyclone.

Additional criteria were tested in order to exclude false cases, such as systems close to land or the remaining few extra-tropical cyclones. However, due to the large variety of the false positives, the additional criteria increase the miss rate, with only limited improvement on the false positive rate.

An estimated miss rate of 48% and a false positive rate of 10 - 20% indicate that the climatology has rather strict criteria which tend to exclude more correct cases than including wrong ones. When considered in the light of disagreement in manually-
detected PL lists, these rates express the high quality of the derived climatology.

## 4   Global polar-low climatology

The PL climatology, consisting of the cyclone tracks that satisfy the four PL criteria, is investigated in this section.

### 4.1   Spatial distribution of polar lows

Figure 3 shows the spatial distribution of the annual mean PL activity between 1979 and 2020. The PL activity is measured
by the number of PL time steps, those satisfying all four PL criteria simultaneously, within a distance of 200 km, which is





approximately the mean radius of PLs (Blechschmidt, 2008; Rojo et al., 2015). Hence, this measure of the PL activity estimates the time a location is affected by a PL.

The spatial distribution of the PL activity agrees in many aspects with the climatologies derived from ERA-I and ASR by Stoll et al. (2018) and with climatologies derived from dynamical downscaling of the NCEP reanalysis for the North Atlantic
(Zahn and von Storch, 2008a) and the North Pacific (Chen and von Storch, 2013). This demonstrates the robustness of PL climatologies derived from reanalysis datasets independent of the specific dataset, tracking algorithm, or PL criteria. However, also some improvements are recognized in this climatology.

PLs are observed in all ocean basins at high latitudes. In general, the highest PL activity is found at some 100 km distance from land masses or the sea-ice edge and decays towards the open sea. In the North Atlantic, most PLs appear North of
50° N. In the North-West Pacific, considerable PL activity is recognized until 40° N. More PLs develop in the Western than the Eastern Pacific. The most equator-ward PLs are found in the Sea of Japan at around 35° N. In the Southern Hemisphere, most PLs develop in the latitude band between 50° and 65° S.

Increased PL activity occurs in ocean basins semi-enclosed by land or sea ice. Accordingly, the highest density of PLs is in the Nordic Seas, especially between Norway and the Svalbard archipelago with an average PL activity of 4 days per year. This
part of the Nordic Seas is known for vigorous PL activity by operational meteorologists (Noer et al., 2011). The second-highest density of PLs is found in the Irminger Sea between Greenland and Iceland up to 3 days per year. The Iceland Sea, to the North of Iceland, in between the two most active PL regions, features rather little PL activity of less than 1 day per year. In the Pacific, the Western Bering Sea is the region of highest PL activity with 2 days per year. Also, a high density of PLs is found in the Sea of Okhotsk, the Labrador Sea, the Gulf of Alaska, and the Sea of Japan with 1 - 2 days per year. All these basins are known for
regular PL activity.

The climatology includes some PLs in the Hudson Bay, in marginal basins of the Arctic Ocean, the Kara, Laptev, and Chuckchi Sea, and in basins not considered being in the polar climate zone, the Mediterranean and the Black Sea. The PL density in these basins is a few hours per year, which means that PLs influence a given location approximately once per decade. These basins are known for their occasional appearance of PLs or related cyclones. In the Hudson Bay, a PL was
observed in 1988 (Gachon et al., 2003). In the Chuckchi Sea, polar meso-scale cyclones were observed in the open waters of the freezing season (Pichugin et al., 2019). The included PLs in the Mediterranean and the Black Sea are likely medicanes, the Mediterranean sibling of PLs. However, the tracking and detection algorithm of this study are not tuned for detecting medicanes and may not capture all of them.

In the Southern Hemisphere, the PL activity is mainly within a latitude band 65° -50° S in the vicinity of the sea-ice cover of
Antarctica and decays towards more temperate latitudes. The Amundsen Sea in the South-Eastern Pacific and a region south of New Zealand in the South-Western Pacific feature the highest PL density with more than 1 day per year. Generally, the density is lower in the Southern Hemisphere than in its northern counterpart. However, due to a larger ocean area, the total activity is only one-third (37%) lower. The regions with increased PL activity of the Southern Hemisphere are similar to the areas found by Stoll et al. (2018).





### 4.1.1 Area of highest polar-low density

In previous PL climatologies based on reanalysis datasets, the Irminger Sea features the highest density (Zahn et al., 2008; Stoll et al., 2018). However, Golubkin et al. (2021) doubted that the Irminger Sea has a larger PL density than the Nordic Seas by investigation of their manually-derived PL list. Accordingly, this climatology has a higher density of PLs in the Nordic Seas than the Irminger Sea, with the latter being globally the area of second-highest density.

Climatologies of Zahn et al. (2008) and Stoll et al. (2018) likely include a considerable amount of orographically-induced shear zones close to the coast of Greenland. Inspection of cases reveals that the here-presented climatology includes only a few shear zones in the Irminger Sea. This observation is supported by a sensitivity study, with the additional exclusion of PL time steps close to land ($< 200\,\text{km}$), which excludes systems mainly induced by orography. In this sensitivity study, the PL density is reduced in the Irminger Sea, but also for the other ocean basins, such that the Irminger Sea remains the area of second-highest PL density (Fig. S 2). Investigation of the additionally excluded cases reveals that a considerable amount of reasonable PL cases are omitted and hence a criterion using the distance to land is not used for the PL climatology.

### 4.1.2 Sensitivity climatologies

Sensitivity climatologies are computed in order to test the dependence of the climatology with regard to the threshold in the PL criteria. Four climatologies are derived by making one of the PL criteria more strict and whereas the other three PL criteria remain the same as for the PL climatology. The strict criteria is determined, such that it is satisfied by 70 - 80% of the PLs from the lists by Noer, Rojo, Yanase and Golubkin, instead of 90% of all lists (see Table S 2).

For all of the stricter criteria, the fraction of additionally excluded cyclone tracks is higher than that of additional excluded PLs from the lists. Hence, the stricter criteria appear successful in the exclusion of borderline cases. Especially increasing the static-stability and the intensity threshold leads to a large exclusion of additional cyclones.

Generally, the sensitivity climatologies (Fig. S 3, 4) reveal similar spatial distributions to the PL climatology (Fig. 3) with the same areas featuring high activity. This demonstrates that the PL climatology is independent of the exact threshold in one of the criteria. However, also some differences in the spatial distributions appear and are shortly discussed in the supplement. This points toward some local differences in the PLs, which is further investigated in Section 5.

### 4.2 Seasonal distribution

The PL climatology, derived based on all months of the year, is able to capture the seasonal distribution (Fig. 4) known from observational studies (e.g Noer et al., 2011). The reproduction of the seasonality is a good test for the quality of a PL climatology.

The displayed seasonal distribution has a large similarity with the one from Stoll et al. (2018) and Chen and von Storch (2013). Generally, PLs develop in the extended winter season of each hemisphere. In the Northern Hemisphere, most PLs develop between November and March with a maximum in December and January with a monthly activity of 37 and 39 days, respectively, where PLs occurring simultaneously count individually into the activity. February has a slightly lower activity





of 32 days, November and March of approximately 21 and 23 days, respectively. Few PLs develop in October and April with around 7 days of activity per month, and in May and September PL development is seldom (1 day per month). In the summer months of June - August PLs are not developing.

In the Southern Hemisphere, the main PL season is between May and November with the highest activity in July (23 days), August (22 days) and June (22 days), and a bit lower activity in September (19 days) and May (18 days). Hence, the winter season features considerably fewer PLs in the Southern than the Northern Hemisphere. In the transition season, October and April, the PL activity is slightly higher in the Southern (11 days), than in the Northern Hemisphere (7 days). The Southern Ocean features a few PLs in March and November (4 days), considerably more than for the Northern Hemisphere corresponding months of May and September. For the summer months, the PL activity is also low in the Southern Hemisphere with less than 1 day per month, but PLs can occur. In conclusion, the PL activity is weaker in the Southern Hemisphere, less constrained to the winter, and more spread throughout the year.

The seasonal distribution varies across the ocean sub-basins (Fig. 5). The Nordic Seas and the Irminger Sea have the longest PL seasons with a long period of high activity from December to March, considerable activity in November and for the Nordic Seas also in April. Both regions have their maximum activity in January, which is more pronounced for the Irminger Sea. For the Nordic Seas, the seasonal distribution is in good agreement with the distributions derived from manual-derived PL lists of Noer et al. (2011) and Rojo et al. (2015). The season is also long in the Gulf of Alaska ranging from November to April with maximum activity in December and March, and fewer PLs in January. In the Labrador Sea and the Bering Sea, most PLs develop from December to February, and some in November and March, meaning that PLs are more constraint to the winter.

In the Sea of Okhotsk and the Sea of Japan, the PL season is shortest with two months containing most PLs, and very little activity from April to October. In the Sea of Okhotsk, most activity is in December and January. From February into the spring, the activity is reduced, likely due to sea ice that covers parts of that area. In the Sea of Japan, January and February are the most active months, probably since this is the time when cold-air advection from the Eurasian continent has the lowest temperatures.

In the Southern Ocean, the peak in PL activity is different for the ocean basins. In the South-East Pacific and the Indian Ocean, most activity occurs in August and the season is slightly shifted towards the later winter, whereas the South-West Pacific and Atlantic feature the highest activity in June and an earlier season.

### 4.3 Time series, trend, inter-annual variability

The average time of PL activity is 172 days per year for the Northern and 137 days for the Southern Hemisphere, with an inter-annual variability of 22 and 15 days, respectively (Fig. 6). Note, that it is common that multiple PLs occur simultaneously, where each center counts to the PL activity.

In the Southern Hemisphere, the PL activity is rather constant during the investigated period from 1979 to 2020, also for each of the ocean basins. Differently, in the Northern Hemisphere, a significant (p-value: 0.005), positive linear trend in the PL activity of 7.6 days per decade is observed. The largest contribution is from the Nordic Sea and the Labrador Sea, an increase of 4.2 and 1.9 days per decade, respectively. The positive trend in PL activity is also observed for the sensitivity climatologies, pointing towards the trend being robust for the choice of PL criteria. The trend is significant for the sensitivity climatologies,



except for the PLs occurring at low static stability, where the trend is also weaker. This points towards that the choice in the threshold for the static stability may influence observed trends in PL activity.

A comparison to the time series presented in Stoll et al. (2018) reveals that years of high and low PL activity are sometimes in disagreement. Also, the observed trends are different, a decrease in intense PLs as observed by Stoll et al. (2018) is not reproduced. It appears that the observed trends in PL activity should be treated with some caution.

## 5 Differences between polar lows

In this section, differences among the PLs from the ocean basins and for the environmental vertical wind-shear categories are investigated.

### 5.1 Parameter comparison in the different ocean basins

Generally, PLs from the different ocean sub-basins share many characteristics, expressed by rather similar parameters (Fig. 7), although some differences are apparent. The specific structure and location of an ocean-sub basin can lead to different typical environmental conditions which influence the PL development. Likely also ocean currents are influencing PL environments in the different ocean basins, but this is not investigated here.

Most PLs occur in close vicinity to land, especially in the Northern Hemisphere ($\approx 500\,\text{km}$). This is less the case for the Southern Ocean since the sea ice around Antarctica creates a buffer between the continent and the open water in which PLs develop. The spatial distribution of PLs for the Southern Hemisphere (Fig. 3b) features the highest density of PLs in the vicinity of the climatological sea ice edge. In the SW Pacific, the typical distance to land is lower than for the rest of the Southern Ocean, due to the presence of multiple smaller islands. In the Sea of Japan, all PLs occur close to land ($< 300\,\text{km}$) since it is bounded by continents and islands. Also in the Nordic Seas, the Labrador Sea, and the Sea of Okhotsk, semi-enclosed by continents and islands, the typical distance to land for PLs is around $300\,\text{km}$. For basins of the Northern Hemisphere, most PLs develop at larger distances (median $\approx 500\,\text{km}$), but still in the vicinity to land. This indicates that the fetch of the air masses in which PLs develop is often of only a few $100\,\text{km}$.

The typical vortex diameter of PLs is around $300\,\text{km}$ (Fig. 7). PLs in the Southern Hemisphere (median: $320\,\text{km}$) and the Irminger Sea ($310\,\text{km}$) are larger, whereas they are smaller in the Sea of Japan ($255\,\text{km}$). These size differences are explained by differences in the typical distance to land, which can mask the vortex area of a PL.

The PLs in the climatology have a typical intensity of $\xi_{smth,850hPa} = 27 \times 10^{-5}\text{s}^{-1}$. They are slightly weaker in the Southern Hemisphere ($25 \times 10^{-5}\text{s}^{-1}$), and stronger in the Sea of Japan ($30 \times 10^{-5}\text{s}^{-1}$). Again, this appears to be an artifact by the vortex area being masked and that large vortices tend to have a higher smoothed relative vorticity.

PLs have a typical lifetime of $20\,\text{hours}$ with $50\%$ between 12 and $32\,\text{hours}$. PLs are often of a longer lifetime in open ocean basins, such as the Indian and Atlantic Ocean, the SE Pacific, and the Bering Sea. Whereas, they are of a shorter lifetime in the Sea of Japan, surrounded by land. Since PLs are slowly decaying after encountering landfall, the duration is shorter in ocean basins bounded by land.





The typical potential temperature at the tropopause in the PL environments is between 290 and 295 K. The median is lower for the Labrador Sea (288 K) and the Sea of Okhotsk (291 K). These ocean basins have continental landmasses to the west and
560 are semi-enclosed by land, such that the marine influence is small and they feature conditions deep in the polar air masses, at low tropopause temperatures.

The potential static stability in PL environments, $\theta_{500hPa} - \theta_{SST}$, is typically around 7 K. PLs in the Southern Ocean and the Gulf of Alaska are featuring a slightly higher (8.5 K), and PLs in the Labrador Sea and Sea of Japan lower static stability (6 K). When the static stability is measure by $SST - T_{500hPa}$, PLs in the Northern Hemisphere feature a typical value of around
565 44 K, whereas it is lower at 42 K for PLs in the Southern Hemisphere. The difference between the hemispheres is larger when using $SST - T_{500hPa}$, than $\theta_{500hPa} - \theta_{SST}$, due to generally lower sea-level pressure in the Southern Ocean at winter times, than anywhere else.

The lifetime-maximum in the vertical wind-shear strength in PL environments is mainly between 2 and $3 \times 10^{-3} \mathrm{s}^{-1}$. The shear is considerably higher for the Sea of Japan, where it has a typical value of $4 \times 10^{-3} \mathrm{s}^{-1}$, and slightly higher for the Sea of
570 Okhotsk and the Labrador Sea ($3 \times 10^{-3} \mathrm{s}^{-1}$), and lower in the Gulf of Alaska, the South Eastern Pacific, and the Indian Ocean. The next section presents that regions with a higher shear strength feature more forward-shear PLs, and regions with a lower shear strength more weak-shear PLs.

## 5.2   Shear distribution in the different ocean basins

The five different shear categories specified by Stoll et al. (2021) appear in every ocean basin. For both hemispheres, and
575 all defined ocean-sub basins, forward-shear PLs are the most common (Fig. 8, 9), responsible for a third of the time steps. (33% in both hemispheres). Approximately a quarter of the time steps are of weak shear (21% Northern and 26% Southern Hemisphere). A slightly lower fraction is of left shear in the Northern (18%) and right shear in the Southern Hemisphere (25%), situations where PLs propagate towards warmer environments. Reverse-shear PLs are more common in the Northern (14%), than in the Southern Hemisphere (7%). Also, PLs propagating into colder environments are more frequent in the Northern
(right shear: 15%), than in the Southern Hemisphere (left shear: 6%).

Some differences are apparent for the ocean sub-basins. Reverse-shear PLs are more common in the Labrador Sea (16% of the situations in the basin), Irminger Sea (16%), and Nordic Seas (14%), and rather seldom in the Gulf of Alaska (8%). Forward-shear situations are responsible for a larger fraction in the Sea of Japan (40%) and the Sea of Okhotsk (39%) and a smaller fraction in the Irminger Sea (25%). Weak shear is more common in the Gulf of Alaska (33%) and less common in the
585 Sea of Japan (8%), Labrador Sea (13%), and Sea of Okhotsk (14%). The highest fraction of left-shear PLs occurs in the Sea of Japan (27%) and the Labrador Sea (20%), likely due to these ocean basins being bounded by land to the West and East which favors southerly, warm-ward flow.

For the Southern Hemisphere, the most notable difference across the sub-basins is that weak-shear PLs are more common in the South-East Pacific and the Indian Ocean than in the South-West Pacific and Atlantic, while in the latter two regions,
right-shear situations are more frequent than in the former two.



## 5.3 Differences in the shear categories

As for PLs in the different ocean sub-basins, PLs associated with the different shear environments share many characteristics (Fig. S 5), however, some differences occur (Fig. 10). PLs in forward shear are characterized by considerably stronger shear strength than those in left, right, and reverse shear. Weak-shear systems are by definition of weaker shear strength. The potential static stability, $\theta_{500hPa} - \theta_{SST}$, is slightly lower for reverse-shear PLs (median in the Northern Hemisphere: 7.0 K) and PLs propagating towards warmer environments (6.8 K), than for forward shear (7.6 K) and PLs propagating towards colder environments (7.5 K). The lower potential static stability can offset the weaker baroclinic growth rate induced by the lower shear strength of these systems.

PLs in environments of weak shear feature similar static stability as the mean of the other strong-shear classes. The intensity of PLs in a situation of weak shear is slightly lower than of the other shear categories, and weak shear occurs mainly at later stages in the life cycle of PLs. Both these aspects point toward that PLs in a weak shear being mainly in the decaying stage in accordance to Stoll et al. (2021).

## 6 Discussion and conclusion

This study presents a new climatology of PLs based on the ERA-5 reanalysis for the years 1979 - 2020. The criteria for the detection of PLs are derived from a comparison of five PL lists and one mesocyclone list from the literature to meso-scale cyclone tracks derived by a tracking algorithm. The following set of criteria is successful for the identification of PLs: (1) Polar-front criterion: $\theta_{trop} < 300.8$ K, (2) Static-stability criterion: $\theta_{500hPa} - \theta_{SST} < 11.0$ K, (3) Intensity criterion: $\xi_{smth,850hPa} > 20.0 \times 10^{-5} s^{-1}$, and (4) Mesoscale-size criterion: Vortex diameter $< 430$ km. These four PL criteria capture the characteristics generally associated to PLs.

The transition between PLs and other types of cyclones is seamless in multiple aspects: the air mass of occurrence, the intensity, and the size. Hence, a specific PL definition is not generally accepted yet (Moreno-Ibáñez et al., 2021) and the choice whether a specific phenomenon is labeled as PL can be subjective. However, agreement appears to exist that PLs are extremes within the large variety of cyclones, and therefore deserve special attention. The lack of a definition complicates the research of PLs. The here-presented PL criteria, derived by comparison of different PL lists by considering the characteristics associated with PLs, may provide a step towards a specific PL definition. The criteria are generally applicable to gridded datasets and robust towards variations in the thresholds.

The derived PL climatology is trustworthy for multiple reasons: (i) Estimated miss and false-positive rates are reasonable compared to the disagreement of observational PL list. (ii) Without constraints, the detected PLs satisfy known characteristics, such as a typical lifetime of 15 - 30 hours and a near-surface wind speed larger than $15 \, \mathrm{m\,s^{-1}}$. (iii) The PL climatology captures know seasonal distributions for individual ocean sub-basins. PLs are winter phenomena, whereas the length of the PL season is considerably different depending on the ocean basin. (iv) For all ocean sub-basins, the PL climatology reproduces spatial distributions in good agreement with observational literature. PLs develop in all ocean basins at high latitude, preferred in close vicinity to land masses or sea ice, in agreement with previous manually and objectively-derived PL datasets. Different





to previous climatologies based on reanalysis datasets (e.g. Stoll et al., 2018; Zahn and von Storch, 2008a), the Nordic Seas
features globally the highest density of PLs, in agreement with the observational dataset of Golubkin et al. (2021).

For the Northern Hemisphere the climatology captures a significant, positive trend of PL activity, mainly due to an increase
in the Nordic Seas and the Labrador Sea. This is different from other climatological studies that observed constant PL activity
for the North Atlantic (Stoll et al., 2018; Zahn and von Storch, 2008a), and climate model projections that predict decreasing
PL activity for a future warmer climate. However, Chen and von Storch (2013) observes a slightly increasing trend of PLs
for the North Pacific, using a weaker static-stability criterion. Also, this study uses a rather weak static-stability criterion. The
application of a stricter static-stability criterion makes the increase in PL activity become statistically non-significant, pointing
towards sensitivity in the PL trend depending on the choice in the static-stability criterion.

This study compares, for the first time, the characteristics of PLs in the different ocean sub-basins. In general, PLs share many
characteristics independent of their ocean basins of development, where differences are explained by the specific configurations
of the basin. For example, PLs in the Sea of Japan and the Labrador Sea develop more often in environments of lower static
stability, whereas PLs in the Southern Ocean and the Gulf of Alaska occur in slightly more stable environments. In the former
regions, the upstream air is often influenced by a long fetch over the winter-time cold continent, hence the air mass has time to
cool, whereas the latter region is more maritime influenced. Generally, the variability of PLs within each basin is larger than
the difference between PLs of the different basins.

This study also investigates globally the fraction of PLs in different vertical wind-shear environments. The five vertical
wind-shear categories defined by Stoll et al. (2021) occur in all regions of PL activity. Forward-shear PLs are the most common
everywhere, making between 25% (Irminger Sea) to 41% of the PL time steps. PLs in weak-shear environments occur in 22%
and 26% of the time for the Northern and Southern Hemisphere, respectively. Weak-shear situations are seldom in the Sea of
Japan, the Sea of Okhotsk, and the Labrador Sea but rather frequent in the Gulf of Alaska, the Indian Ocean, and the SE Pacific.

PLs propagating towards warmer environments, being of left shear in the Northern (18%) and right-shear in the Southern
Hemisphere (25%), are also quite common. Reverse-shear PLs are rather seldom, however, more common in the Northern
(14%) than the Southern Hemisphere (7%). PLs propagating towards colder environments are of approximately similar oc-
currence to reverse-shear PLs in both hemispheres. Also for these two shear types, considerable regional differences in the
frequency exist. They have the highest shares in the North Atlantic and the Sea of Okhotsk.

Most environmental parameters are quite similar for the shear categories, and differences are in accordance to Stoll et al.
(2021). The shear strength is often larger for forward-shear PLs, whereas the static stability is often lower for reverse-shear
PLs and the PLs propagating towards warmer environments. Hence, the contributors to a high baroclinic growth rate are
slightly different within the strong-shear categories. Weak-shear situations are mainly occurring in the decaying phase of the
PL, pointing towards that the intensification of PLs requires environmental baroclinicity.

*Data availability.* The polar-low climatology is provided.





*Acknowledgements.* Thanks to ECMWF for providing access to data from the ERA-5 reanalysis. The data were processed at the super-computer FRAM and stored at NIRD, both provided by the Norwegian Metacenter for Computational Science (NOTUR) under the projects NN9348K and NS9063K, respectively.



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



**Figure 2.** As Figure 1, the distribution in different parameters for all tracks of the Northern and Southern Hemisphere (black), different polar low lists (colors) and in the tracks that remain after application of the polar-low criteria for the two hemispheres (black dotted and dash-dotted). (a) The maximum wind speed at the tropopause poleward of the system, $U_{trop,polew}$, is computed as the lifetime mean of the track. (b) The temperature difference between the 500 hPa level and the sea surface, $SST-T_{500hPa}$, and (d) the maximum near-surface wind speed, $U_{10m}$, as lifetime maximum. (c) The potential temperature difference between 500 hPa and 925 hPa, $\theta_{500hPa} - \theta_{925hPa}$, and the sea-level pressure, as lifetime minimum.

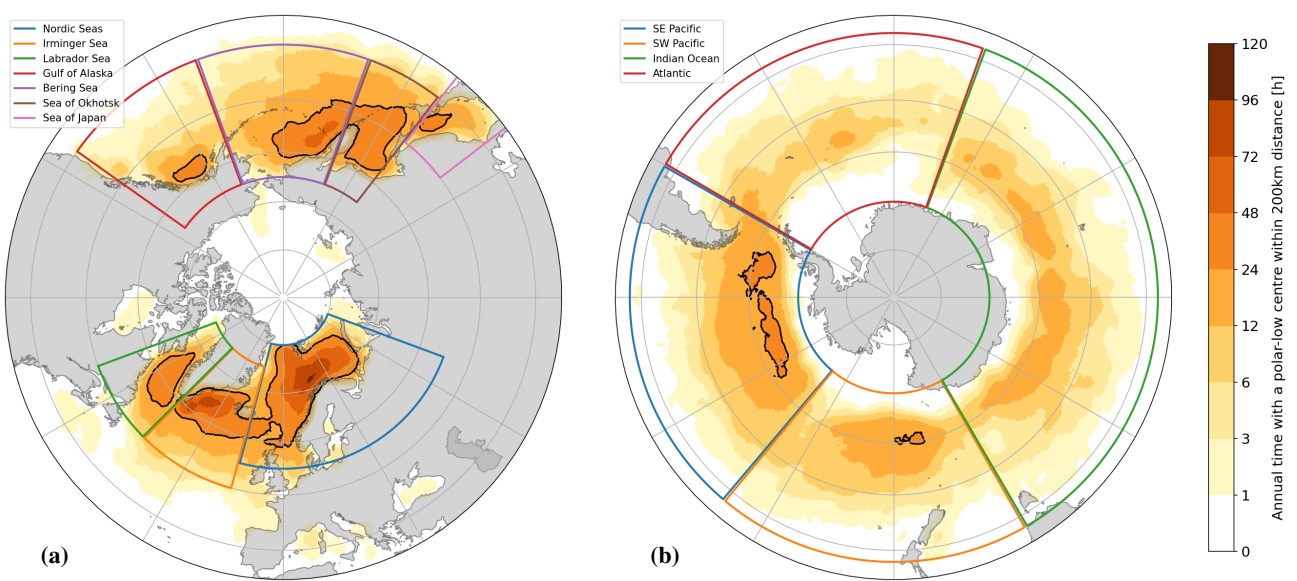

**Figure 3.** Mean annual density of polar-low activity, measured by the amount of polar-low time steps within a 200 km distance, which is a typical radius of a polar low, for (a) the Northern and (b) the Southern Hemisphere. Note the logarithmic color scale towards low densities. A black contour encircles regions with an activity of more than 24 h per year, which means that a given location is affected by a polar low with a typical duration of one day approximately once per year.

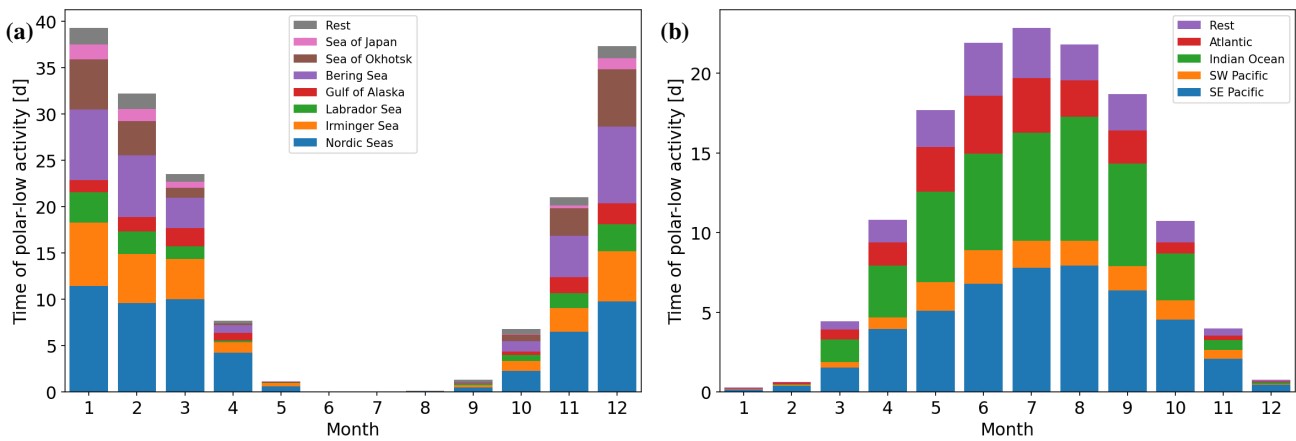

**Figure 4.** The average monthly time of polar-low activity normalized to months with a duration of 30 days for (a) the Northern and (b) the Southern Hemisphere. Note that two polar lows occurring simultaneously count twice to the activity. Colors denote the contribution of specific regions marked by boxes in Figure 3.



**Figure 5.** The average monthly time of polar-low activity normalized to months with a duration of 30 days for the specific regions marked by boxes in Figure 3.



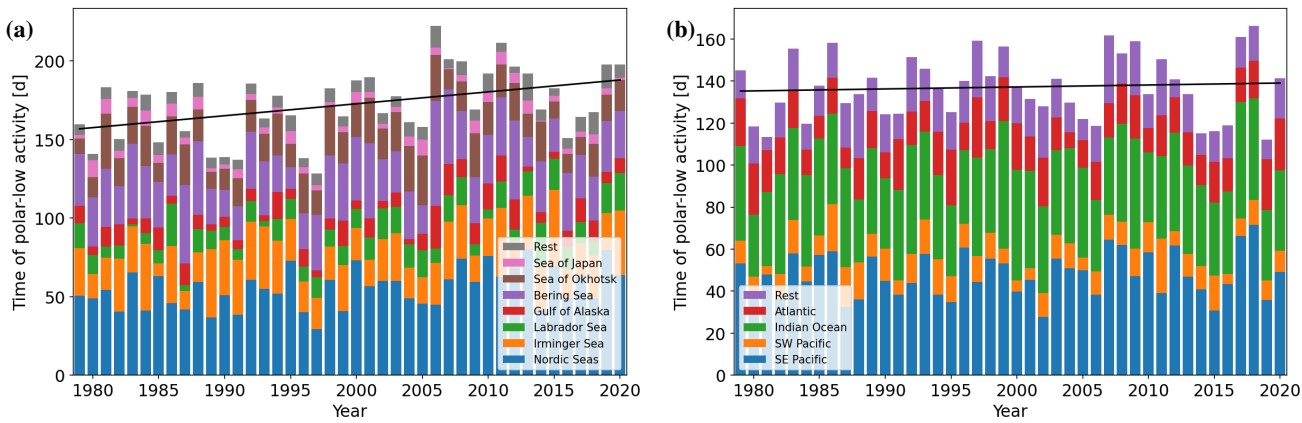

**Figure 6.** The annual time of polar-low activity for (a) the Northern and (b) the Southern Hemisphere. Colors denote the contribution of specific regions marked by boxes in Figure 3.



**Figure 7.** Parameter distributions polar lows from the different regions. For (a) the distance to land, (b) the vortex diameter, (c) the relative vorticity, (g) $SST - T_{500hPa}$, and (h) the shear strength the lifetime-maximum of the PL tracks is computed. For (e) $\theta_{trop}$ the lifetime-mean and for (f) $\theta_{500hPa} - \theta_{SST}$ the lifetime-minimum.





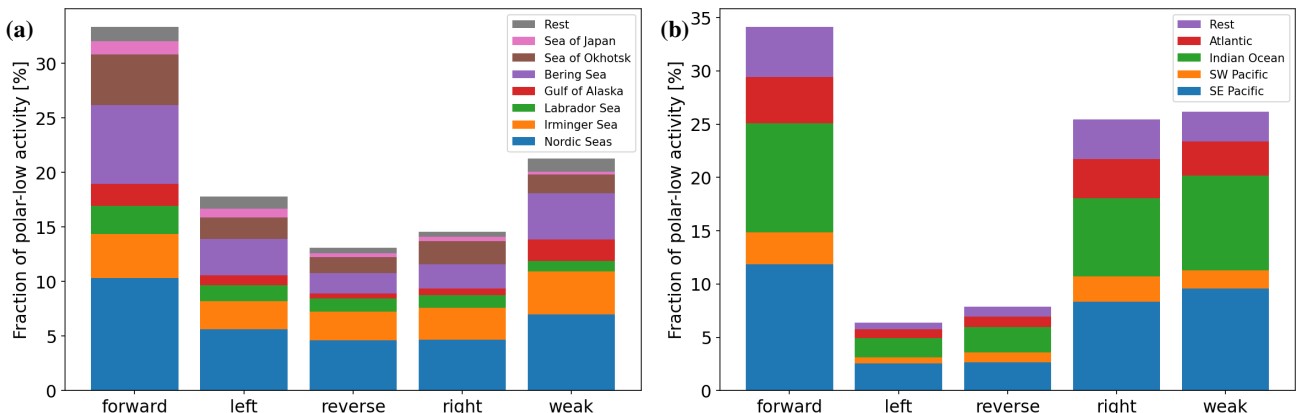

**Figure 8.** Distribution of the different shear categories in the (a) Northern, and (b) Southern Hemisphere.

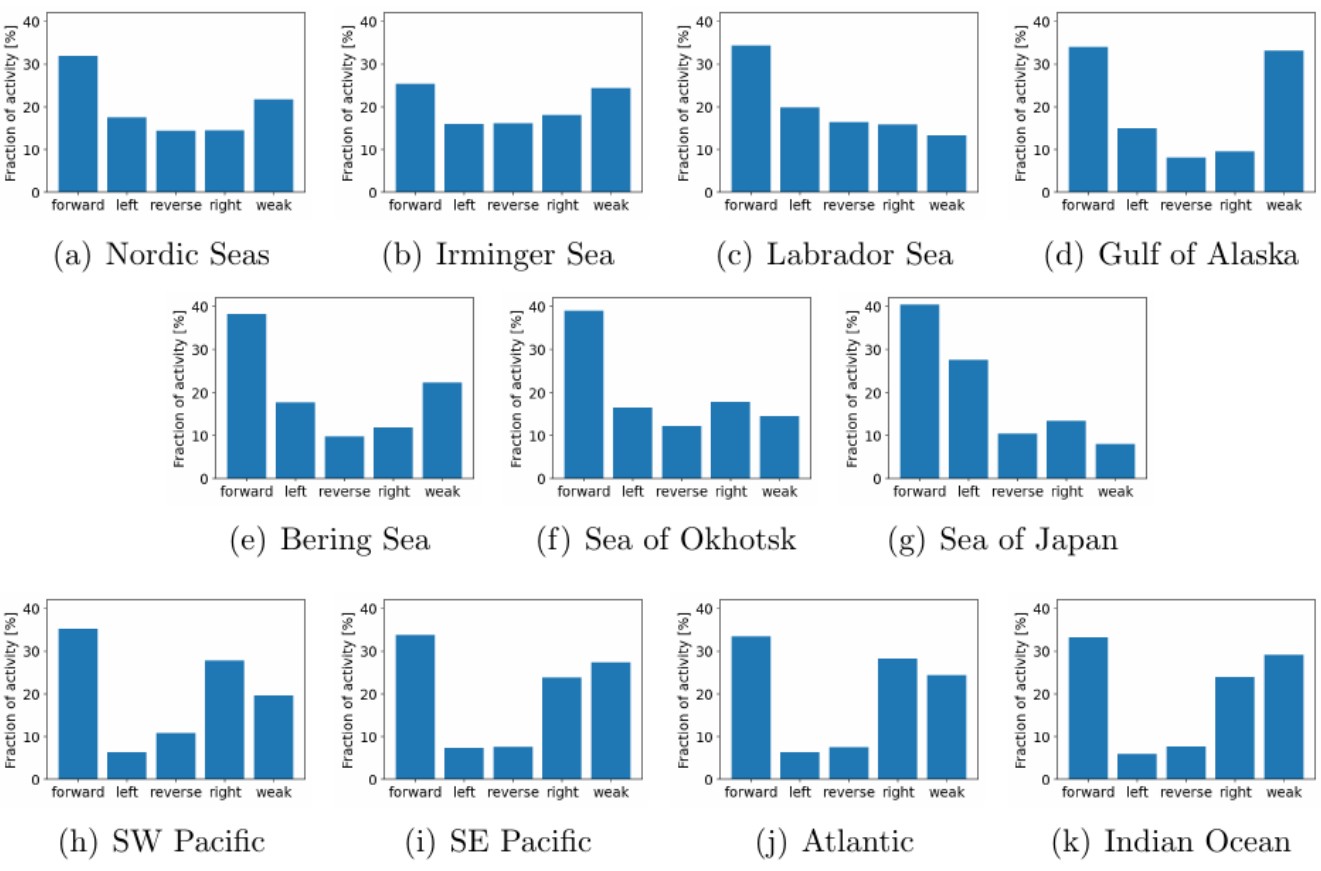

**Figure 9.** The average monthly time of polar-low activity normalized to months with a duration of 30 days for the specific regions marked by boxes in Figure 3.



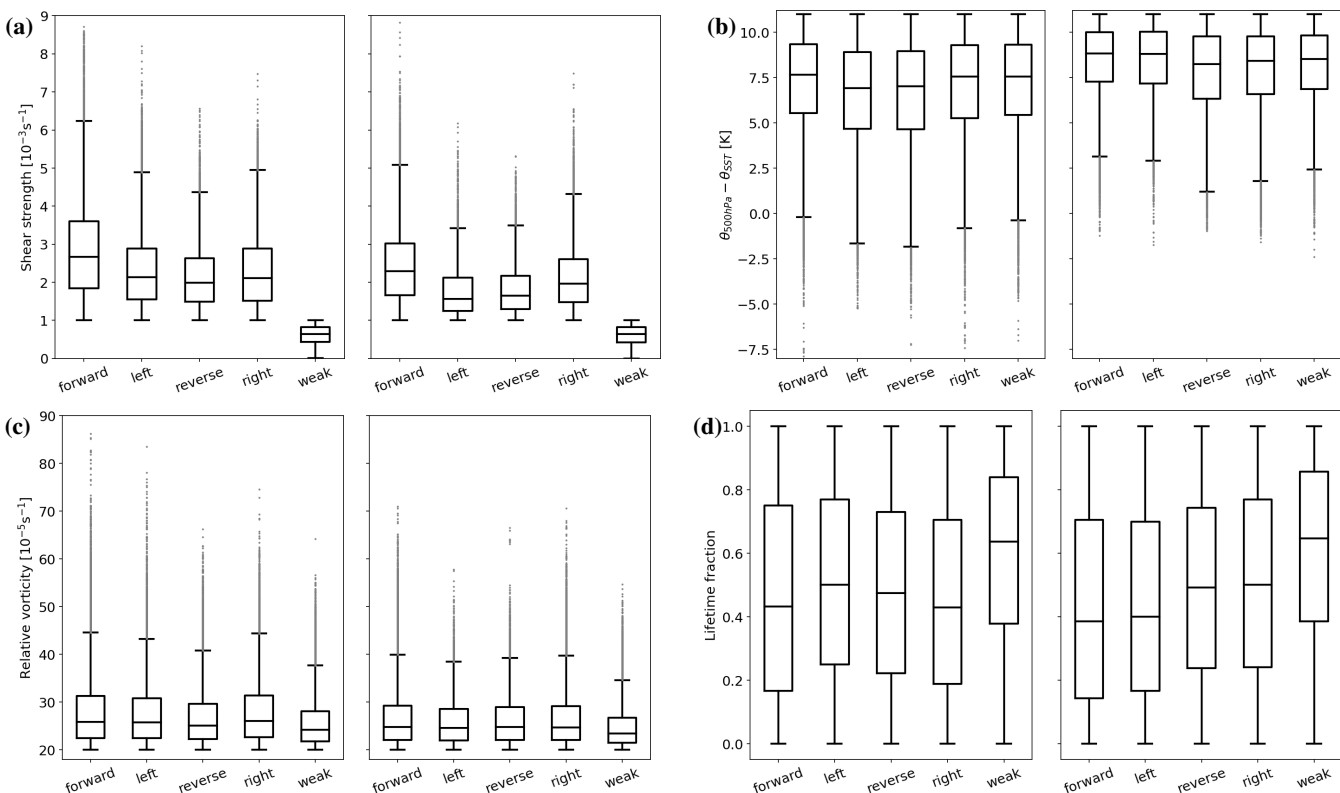

**Figure 10.** Parameter distribution in (a) the vertical wind-shear strength, (b) the static stability, (c) the relative vorticity, and (d) the lifetime fraction for the different shear categories of (left) the Northern and (right) the Southern Hemisphere. The parameters are computed for all polar-low time steps within the shear category, since a given polar low is often changing shear category through its life. Note that left shear in the Northern Hemisphere and right shear in the Southern Hemisphere is attributed to PLs propagating towards warmer environments.