# Peer review of "A global climatology of polar lows investigated for local differences and wind-shear environments"

_Weather and Climate Dynamics, 2021_

## Referee Comment (RC1)

**REFEREE COMMENT**

Title: A global climatology of polar lows investigated for local differences and wind-shear environments
Author: Patrick Johannes Stoll

**1. GENERAL COMMENTS**

Most of the climatologies of polar lows (PLs) focus on a particular region, mainly in the Northern Hemisphere. A previous work of the same author provided the first much-needed global climatology of PLs using the predecessor of ERA5 and a regional reanalysis. This new study makes a significant contribution to the study of PLs by providing a global climatology of PLs based on the recently released ERA5 reanalysis.

The methodology used is adequate, and the selection of parameters to detect PLs is reasonable and well explained. The analysis of the climatology focuses on the main points of interest for the research community. In particular, the focus on the differences between PLs from different ocean basins is a subject that has received little attention in the past; thus, this is also an important contribution of this work. Finally, the section "Validation: Misses and False Positives" highlights one of the main problems of developing the climatology of PLs, which is selecting the adequate PL criteria that will minimize false positives and misses.

In general, the text is well-structured and easy to follow. The language is clear and concise, although there are several technical corrections that need to be done. Some minor corrections are also needed.

**2. SPECIFIC COMMENTS**

*1. Introduction*

- p. 1, line 25, & p.3, line 81: The author uses Hersbach and Dee (2016) as reference, but that is a short text that was part of ECMWF's newsletter that was written before the release of ERA5. Therefore, I recommend using Hersbach et al. (2020) since it describes the ERA5 reanalysis in detail:

  Hersbach, H., Bell, B., Berrisford, P., Hirahara, S., Horányi, A. and co-authors. 2020. The ERA5 global reanalysis. Q. J. Roy. Meteorol. Soc. 146, 1999-2049. doi: 10.1002/qj.3803

- p. 2, lines 49 – 51: There should be a reference for those two sentences. For the sentence "Criteria for these characteristics are to some degree arbitrary due to a smooth transition between PLs and other cyclones", Yanase et al. (2016) would be a good reference.

*2. Data and methods*

- p.6, lines 154-155: The Verezemskaya et al. (2017) study did analyze whether a system was intense enough to be considered as PL. Verezemskaya et al. (2017) found 1735 mesocyclones and classified three quarters of those to be PLs. Although the criteria they used to classify mesocyclones as PLs was not very strict (mesocyclones developing over the ocean with maximum wind speeds > 15 m/s at least once during their lifetime), this reduced list would be more appropriate for this study than the whole list of polar mesocyclones, and it could potentially contribute to the parameter derivation of this study.
  If the Verezemskaya list does not contribute to the parameter derivation of this study, it does not seem necessary to include it (and it is a little bit confusing for the reader to see that list in the study). It would be more appropriate to show the results obtained using the Verezemskaya list in the supplementary material.

- p.7, line 195: Since there is no universal agreement on the PV threshold to define the dynamical tropopause, it would be relevant to add a reference here for the choice of 2 PVU.

*3. Polar-low criteria*

- p. 8, lines 217 – 221: I suggest moving the summary of the PL criteria to the end of this section (before section 3.1), after the author has explained how the thresholds for the parameters have been selected. Otherwise, the reader gets the impression that the PL criteria have been obtained from another study.

- p.8, line 236-237: This sentence is not clear. The legend of Table 3 explains this a little bit better (it says: ""The last column displays the additionally excluded cyclones by the threshold after application of the polar-low criteria of different type.").  Since the other types of polar low criteria include different thresholds, it is not clear which thresholds are being considered here for each of those three criteria.

- p.15, lines 370-373: The statement that PLs with a longer lifetime are better simulated by ERA5 needs a reference to support it.

- p.16, line 420: Rather than "high quality", it seems more appropriate to say that the derived climatology is "reasonable" given the disagreement in manually detected PLs. This section shows how difficult is to obtain a climatology of high quality, and the choice of the author seems reasonable given those challenges. This choice ensures that the climatology corresponds almost only to PLs (given that the false positive rate is 10 - 20%), but may not be complete since all PLs may not be captured (given that the miss rate is 48%).

*4. Global polar-low climatology*

- p.17, lines 451-452: It would be relevant to shortly explain why the medicanes are "the Mediterranean sibling of PLs" and add a reference to support that statement. Some suggested references:

  Businger, S. and Reed, R. J. 1989. Cyclogenesis in cold air masses. Wea. Forecasting 4, 133–156. doi:10.1175/1520-0434(1989)004<0133:CICAM>2.0.CO;2

  Romero, R. and Emanuel, K. 2017. Climate change and hurricane-like extratropical cyclones: projections for North Atlantic polar lows and medicanes based on CMIP5 models. J. Clim. 30, 279–299. doi:10.1175/JCLI-D-16-0255.1

  Moreno-Ibáñez, M., Laprise, R., and Gachon, P.: Recent advances in polar low research: current knowledge, challenges and future perspectives, Tellus A: Dynamic Meteorology and Oceanography, 73, 1–31, 2021.

- p.18, lines 481-482: It is more appropriate to say that this demonstrates that the PL climatology is "not strongly dependent on" the exact threshold in one of the criteria.

- p.19, line 495: According to Figure 4, the main PL season is between April and October, both included.

- p.19, lines 512-513: A reference is needed for this sentence "January and February (…) this is the time when cold-air advection from the Eurasian continent has the lowest temperatures".

*5. Differences between polar lows*

- Section 5.1 (p.20-21): For each parameter, it would be pertinent to write the reference to the figure (e.g., "The typical vortex diameter of PLs (Fig. 3b)").

- p.20, lines 546-547: This does not necessarily mean that "the fetch of the air masses in which PLs develop is often of only a few 100 km" because PLs can develop near a certain landmass (e.g., Svalbard), and travel many km before reaching another landmass (e.g., norther Norway), so the distance to the closest landmass will be of a few 100 km, but the air mass could have actually travelled more km.

- p.21, lines 566-567: A reference is needed for "due to generally lower sea-level pressure in the Southern Ocean at winter times, than anywhere else".

*Figures*

- Figure 5: It is hard to compare the results of the different regions because the *y* axis is different for each region.

**3. TECHNICAL CORRECTIONS**

*Text*

- p.4 lines 99-101: This sentence is hard to understand, so I recommend dividing it into two sentences.
- p. 2, line 43: "intentionally" instead of "intensionally".
- p. 3, line 62: "affected" instead of "effected".
- p. 4, line 103-104: I think this sentence is not needed and does not provide additional information: "they are expected neither over sub-tropical nor over ice-covered areas nor over landmasses."
- p.5, line 124: Add "Hemisphere" after "Northern".
- p.5, line 130: Define the acronym STARS since it is the first time that it is mentioned here: "Sea Surface Temperature and Altimeter Synergy for Improved Forecasting of Polar Lows (STARS)"
- p.7, line 174: "explained by the fact that the list consists of" instead of "explained by the list consist of".
- p. 7, line 175: This sentence is needed because the previous and next sentences are sufficient to explain the results: "The high rate of weak systems in the Verezemskaya list is also expressed by a large number of cases in a short time period of four months as compared to the other PL lists that span multiple years."
- p.8, line 222: "fourth" instead of "forth".
- p.8, line 227: Add "The" before "following".
- p.8, line 229: The fraction of PLs retained is 69-89% according to Table 2. I suppose the author excludes here the results related to the Verezemskaya list.
- p.12, line 286: Add ";" after "(SST−$T_{500hPa}$"
- p.13, line 312: "Col. 6" instead of "Col.5".
- p.13, line 313: "Col. 6" instead of "Col.5".
- p.13, line 314: Consider using other word instead of "punishing".
- p.15, line 371: "explained by the fact that PLs from the lists are biased towards longer lifetimes since" instead of "explained that PLs from the lists being biased towards longer lifetimes, since"
- p.15, line 387: "ground truth" instead of "ground-true".
- p.16, lines 407-408: Add "according to the PL lists" after "whether they are considered PLs".
- p.17, line 454: "edge" instead of "cover".
- p.17, line 456-457: Remove "however" and "only".
- p.19, line 519: Remove ',' after "Note"
- p.20, line 555: "Bering Sea, whereas they" instead of "Bering Sea. Whereas, they"
- p.21, line 564: Add "d" after "measure".
- p.21, line 586-587: A southerly wind is a wind that blows from the south, so I suppose the author means "northerly wind, warm-ward flow".
- p.22, line 601-602: This sentence is not correctly written and does not add additional information, so it could be deleted: "Both these aspects point toward that PLs in a weak shear being mainly in the decaying stage in accordance to".

- p.22, line 611: Write "the classification of a specific phenomenon as PL can be subjective" instead of "the choice whether a specific phenomenon is labeled as PL can be subjective".
- p.22, line 620: "known" instead of "know".

*Tables*

- Table 1, line 2 (legend): Add "list" after "Yanase".
- Table 1, line 6 (legend): There is information missing in this line. I suggest modifying what is written after "the PL from the list," as follows: "and column 7 shows the number of tracks that are excluded since the ERA-5 track matches two PLs from the list".
- Table 2, lines 1-2 (legend): Add "Hemisphere" after "Northern".
- Table 3, lines 3-4 (legend): Delete "in addition also", and add "also" before "presented."
- Table 3, lines 3-4 (legend): Mention which of the rows corresponds to the threshold obtained excluding the Smirnova list.

*Figures*

- Figure 1 (legend): Modify the first sentence because it is not well written, and it is not clear.
- Figure 2, line 5 (legend): Add "(f)" before "the sea-level".
- Figure 2 (legend): The figure *e* needs to be explained in the legend.
- Figure 7 (legend): "Parameter distributions of polar lows for the different regions" instead of "Parameter distributions polar lows from the different regions".
- Figure 7 (legend): The figure *d* needs to be explained in the legend.
- Figure 9 (legend): The quality of the figure needs to be improved.
- Figure 10, line 4 (legend): "are" instead of "is".

---

## Author Response (AR1)

**Response to reviewer 1.**

I thank the referee for the thorough and constructive review. The comments are contributing to an overall improved manuscript.

**SPECIFIC COMMENTS**

Reviewer: *p. 1, line 25, & p.3, line 81: The author uses Hersbach and Dee (2016) as reference, but that is a short text that was part of ECMWF's newsletter that was written before the release of ERA5. Therefore, I recommend using Hersbach et al. (2020) since it describes the ERA5 reanalysis in detail: Hersbach, H., Bell, B., Berrisford, P., Hirahara, S., Horányi, A. and co-authors. 2020. The ERA5 global reanalysis. Q. J. Roy. Meteorol. Soc. 146, 1999-2049. doi:10.1002/qj.3803*

Response: Thanks for the suggestion. The reference is updated accordingly.

Reviewer: *p. 2, lines 49 – 51: There should be a reference for those two sentences. For the sentence "Criteria for these characteristics are to some degree arbitrary due to a smooth transition between PLs and other cyclones", Yanase et al. (2016) would be a good reference.*

Response: Indeed, thanks for the good suggestion.

Reviewer: *p.6, lines 154-155: The Verezemskaya et al. (2017) study did analyze whether a system was intense enough to be considered as PL. Verezemskaya et al. (2017) found 1735 mesocyclones and classified three quarters of those to be PLs. Although the criteria they used to classify mesocyclones as PLs was not very strict (mesocyclones developing over the ocean with maximum wind speeds > 15 m/s at least once during their lifetime), this reduced list would be more appropriate for this study than the whole list of polar mesocyclones, and it could potentially contribute to the parameter derivation of this study. If the Verezemskaya list does not contribute to the parameter derivation of this study, it does not seem necessary to include it (and it is a little bit confusing for the reader to see that list in the study). It would be more appropriate to show the results obtained using the Verezemskaya list in the supplementary material.*

Response: The Verezemskaya list was obtained following the link www.sail.msk.ru/antarctica in the publication. The list does not contain information on the near-surface wind speed obtained in the vicinity of the low, and I find that it is neither ensured that all systems from the list develop poleward of the polar front. I therefore follow the advice of the reviewer and exclude the list from the main manuscript and add it to the supplementary material instead.

Reviewer: *p.7, line 195: Since there is no universal agreement on the PV threshold to define the dynamical tropopause, it would be relevant to add a reference here for*

*the choice of 2 PVU.*

Response: This was added: "The tropopause is defined by the most common choice as the lowest atmospheric level where $2\,\mathrm{PVU}$ ($1\,\mathrm{PVU} = 10^{-6}$ K m$^2$ kg$^{-1}$ s$^{-1}$) is reached (Kunz et al., 2011)."

Reviewer: *p. 8, lines 217 – 221: I suggest moving the summary of the PL criteria to the end of this section (before section 3.1), after the author has explained how the thresholds for the parameters have been selected. Otherwise, the reader gets the impression that the PL criteria have been obtained from another study.*

Response: The beginning of Section 3 has been restructured according to the suggestion.

Reviewer: *p.8, line 236-237: This sentence is not clear. The legend of Table 3 explains this a little bit better (it says: "The last column displays the additionally excluded cyclones by the threshold after application of the polar-low criteria of different type."). Since the other types of polar low criteria include different thresholds, it is not clear which thresholds are being considered here for each of those three criteria.*

Response: L 236ff "The additional value of a criterion is examined by the excluded fraction of cyclones only excluded by this criterion, but not by the three criteria of other types (Col. 6)." was rewritten to:
"The additional value of a criterion is examined by the excluded fraction of cyclones remaining after application of the other polar-low criteria (Col. 6)."
I tried to clarify the mentioned sentence from Table 3 by adding "(in red)", it reads now: "The last column displays the additionally excluded cyclones by the threshold after application of the polar-low criteria (in red) of different type."

Reviewer: *p.15, lines 370-373: The statement that PLs with a longer lifetime are better simulated by ERA5 needs a reference to support it.*

Response: I changed the argumentation for the biases: L370ff "The small difference may be explained that PLs from the lists being biased towards longer lifetimes, since the PLs from the lists require a match in ERA-5, which is more likely the case for PLs with a longer lifetime, which are better simulated by ERA-5 and captured by the tracking and matching procedure." was changed to: "The small difference may be explained that PLs from the lists being biased towards longer lifetimes, since PLs with less than 3 time steps are excluded for assuring a trustworthy track matching."

Reviewer: *p.16, line 420: Rather than "high quality", it seems more appropriate to say that the derived climatology is "reasonable" given the disagreement in manually detected PLs. This section shows how difficult is to obtain a climatology of high quality, and the choice of the author seems reasonable given those challenges. This choice ensures that the climatology corresponds almost only to PLs (given that the false positive rate is 10 - 20%), but may not be complete since all PLs may not be*

*captured (given that the miss rate is 48%).*

Response: This is changed accordingly: "...these rates express that the derived climatology is reasonable."

Reviewer: *p.17, lines 451-452: It would be relevant to shortly explain why the medicanes are "the Mediterranean sibling of PLs" and add a reference to support that statement. Some suggested references: Businger, S. and Reed, R. J. 1989. Cyclogenesis in cold air masses. Wea. Forecasting 4, 133–156. doi:10.1175/1520-0434(1989)004<0133:CICAM>2.0.CO;2 Romero, R. and Emanuel, K. 2017. Climate change and hurricane-like extratropical cyclones: projections for North Atlantic polar lows and medicanes based on CMIP5 models. J. Clim. 30, 279–299. doi:10.1175/JCLI-D-16-0255.1 Moreno-Ibáñez, M., Laprise, R., and Gachon, P.: Recent advances in polar low research: current knowledge, challenges and future perspectives, Tellus A: Dynamic Meteorology and Oceanography, 73, 1–31, 2021.*

Response: Thanks for the suggested references, that discuss some similarities between PLs and medicanes. They are included, but I do not repeat the arguments here.

Reviewer: *p.18, lines 481-482: It is more appropriate to say that this demonstrates that the PL climatology is "not strongly dependent on" the exact threshold in one of the criteria.*

Response: I reformulated: "This demonstrates that the PL climatology is only little dependent on the exact threshold in one of the criteria." This formulation avoids the the negation.

Reviewer: *p.19, line 495: According to Figure 4, the main PL season is between April and October, both included.*

Response: Thanks for spotting the mistake. I corrected it: "In the Southern Hemisphere, the main PL season is from April to October with a maximum in winter (Jun‑Aug; up to 23 days) ..."

Reviewer: *p.19, lines 512-513: A reference is needed for this sentence "January and February (...) this is the time when cold-air advection from the Eurasian continent has the lowest temperatures".*

Response: I have removed the reasoning for the seasonal distribution of the Sea of Japan and rewritten this and the previous paragraph. Further, I have added a paragraph to the end of Section 4.2, that summarizes reasons for the differences in the distributions: "The duration and maximum of the local PL seasons are likely influenced by multiple factors, including (i) the availability of ambient cold air masses upstream of the basin, (ii) the occurrence of synoptic weather patterns advecting the cold air masses over the basin, (iii) the extend of sea ice, (iv) the sea surface

temperature, which is influenced by ocean currents."

Reviewer: *Section 5.1 (p.20-21): For each parameter, it would be pertinent to write the reference to the figure (e.g., "The typical vortex diameter of PLs (Fig. 3b)").*

Response: This was added.

Reviewer: *p.20, lines 546-547: This does not necessarily mean that "the fetch of the air masses in which PLs develop is often of only a few 100 km" because PLs can develop near a certain landmass (e.g., Svalbard), and travel many km before reaching another landmass (e.g., norther Norway), so the distance to the closest landmass will be of a few 100 km, but the air mass could have actually travelled more km.*

Response: I agree and therefore changed the formulation from: "This indicates that the fetch of the air masses in which PLs develop is often of only a few 100 km." to: "The accumulation of PLs in vicinity of land or sea ice and the fast decay of the PL density at distances larger than 1000 km distance from either of the two (Fig. 3) indicates that for long fetches the marine influence is destructing the polar air masses favourable for PL development."

Reviewer: *p.21, lines 566-567: A reference is needed for "due to generally lower sea-level pressure in the Southern Ocean at winter times, than anywhere else".*

Response: I now refer to the ERA-4 atlas by Kållberg et al. (2005).

Reviewer: *Figure 5: It is hard to compare the results of the different regions because the y-axis is different for each region.*

Response: I agree that it is challenging to compare the seasonal distributions and was thinking on how to display the y-axis. One problem is that the regions are of different size, hence that the comparison of absolute numbers is not reasonable. One could suggest to normalize by the area, however the PL density within each region is quite uneven distributed (see Fig. 3), such that the normalization would depend a lot on where the boundary of a box is located, which is unsatisfactory. Therefore, I decided to display the distributions as they are, which is close to compare normalized distributions, however with the y-axis still displaying the total frequency in the box, which is in my opinion relevant information .

**TECHNICAL CORRECTIONS**

Reviewer: *p.4 lines 99-101: This sentence is hard to understand, so I recommend dividing it into two sentences.*

Response: The sentence "At the consecutive time step, the tracking algorithm merges the largest vorticity maxima occurring over open water within 150 km of

an estimated propagation by the mean wind of the 700 and 1000 hPa levels within 200 km distance." was splitted to:
"At the consecutive time step, the tracking algorithm merges the largest vorticity maxima occurring over open water within a distance of 150 km to the point of estimated propagation. The propagation is estimated by the mean wind of the 700 and 1000 hPa levels within 200 km distance."

Reviewer: *p. 2, line 43: "intentionally" instead of "intensionally".*

Response: Thanks for spotting the mistake.

Reviewer: *p. 3, line 62: "affected" instead of "effected".*

Response: Thanks.

Reviewer: *p. 4, line 103-104: I think this sentence is not needed and does not provide additional information: "they are expected neither over sub-tropical nor over ice-covered areas nor over landmasses."*

Response: I agree. "Since PLs develop in marine polar air masses, they are expected neither over sub-tropical nor over ice-covered areas nor over landmasses. Hence, tracks are derived over open water in the latitude band 30° - 80° of both hemispheres." was rewritten to:
"Since PLs develop in marine polar air masses, tracks are derived over open water in the latitude band 30° - 80° of both hemispheres."

Reviewer: *p.5, line 124: Add "Hemisphere" after "Northern".*

Response: Done as recommended.

Reviewer: *p.5, line 130: Define the acronym STARS since it is the first time that it is mentioned here: "Sea Surface Temperature and Altimeter Synergy for Improved Forecasting of Polar Lows (STARS)"*

Response: I added the name of the accrony in the same way as on the website: "STARS (Sea Surface Temperature and Altimeter Synergy for Improved Forecasting of Polar Lows)"

Reviewer: *p.7, line 174: "explained by the fact that the list consists of" instead of "explained by the list consist of".*

Response: Thanks, this was adapted accordingly.

Reviewer: *p. 7, line 175: This sentence is needed because the previous and next sentences are sufficient to explain the results: "The high rate of weak systems in the Verezemskaya list is also expressed by a large number of cases in a short time*

*period of four months as compared to the other PL lists that span multiple years."*

Response: I guess the reviewer meant "This sentence is NOT needed...". Hence, the sentence is removed here and instead in a different formulation added to the presentation of the polar-low lists in Section 2.2.

Reviewer: *p.8, line 222: "fourth" instead of "forth".*

Response: Thanks.

Reviewer: *p.8, line 227: Add "The" before "following".*

Response: Changed accordingly.

Reviewer: *p.8, line 229: The fraction of PLs retained is 69-89% according to Table 2. I suppose the author excludes here the results related to the Verezemskaya list.*

Response: Indeed. This is clarified by adding "he derived set of criteria retains most PLs from the **five** lists (68‑89%)..." and to the caption of Table 2: "Statistics for the satisfaction of the polar-low criteria from the **five** polar low lists, **the mesocyclone list from Verezemskaya,** and all cyclone tracks of the Northern (NH) and Southern Hemisphere (SH)." Also at some other instance it was more clearly separated between the five polar-low lists and the mesocyclone list.

Reviewer: *p.12, line 286: Add ";" after "(SST-$T_{500hPa}$"*

Response: Thanks.

Reviewer: *p.13, line 312: "Col. 6" instead of "Col.5".*

Response: Thanks.

Reviewer: *p.13, line 313: "Col. 6" instead of "Col.5".*

Response: This was removed since the reader should know from the previous sentence where to find the information.

Reviewer: *p.13, line 314: Consider using other word instead of "punishing".*

Response: It was replaced by: "... such that a $SST - T_{500hPa}$ criteria is less frequently satisfied in the Southern than the Northern Hemisphere. "

Reviewer: *p.15, line 371: "explained by the fact that PLs from the lists are biased towards longer lifetimes since" instead of "explained that PLs from the lists being biased towards longer lifetimes, since"*

Response: Thanks.

Reviewer: *p.15, line 387: "ground truth" instead of "ground-true".*

Response: Thanks.

Reviewer: *p.16, lines 407-408: Add "according to the PL lists" after "whether they are considered PLs".*

Response: I think this is a misunderstanding. Hence, I tried to clarify: "For the evaluation of false positives, 100 of the detected PLs were randomly chosen and investigated whether they are considered PLs." was replaced by:
"For the evaluation of false positives, 100 of the PLs **from the derived climatology** were randomly chosen and investigated whether they are considered PLs."

Reviewer: *p.17, line 454: "edge" instead of "cover".*

Response: Thanks.

Reviewer: *p.17, line 456-457: Remove "however" and "only".*

Response: I think the two words make sense to resolve the issue of a lower density versus a higher area to give almost the same amount of PLs. "Generally, the density is considerably lower in the Southern Hemisphere than in its northern counterpart. However, due to a larger ocean area, the total activity is only one-third (37%) lower."

Reviewer: *p.19, line 519: Remove ',' after "Note"*

Response: Thanks.

Reviewer: *p.20, line 555: "Bering Sea, whereas they" instead of "Bering Sea. Whereas, they"*

Response: Thanks.

Reviewer: *p.21, line 564: Add "d" after "measure".*

Response: Thanks.

Reviewer: *p.21, line 586-587: A southerly wind is a wind that blows from the south, so I suppose the author means "northerly wind, warm-ward flow".*

Response: Indeed.

Reviewer: *p.22, line 601-602: This sentence is not correctly written and does not*

add additional information, so it could be deleted: "Both these aspects point toward that PLs in a weak shear being mainly in the decaying stage in accordance to".

Response: I agree that it is not nicely written. Hence, I replaced it with: "Both in accordance to Stoll et al. (2021), who hypothesize that weak-shear situations often associated with spiralli-form clouds are the result of a baroclinic warm-seclusion process."

Reviewer: *p.22, line 611: Write "the classification of a specific phenomenon as PL can be subjective" instead of "the choice whether a specific phenomenon is labeled as PL can be subjective".*

Response: Thanks.

Reviewer: *p.22, line 620: "known" instead of "know".*

Response: Thanks.

Reviewer: *Table 1, line 2 (legend): Add "list" after "Yanase".*

Response: Thanks.

Reviewer: *Table 1, line 6 (legend): There is information missing in this line. I suggest modifying what is written after "the PL from the list," as follows: "and column 7 shows the number of tracks that are excluded since the ERA-5 track matches two PLs from the list".*

Response: Thanks.

Reviewer: *Table 2, lines 1-2 (legend): Add "Hemisphere" after "Northern".*

Response: Thanks.

Reviewer: *Table 3, lines 3-4 (legend): Delete "in addition also", and add "also" before "presented."*

Response: See next comment.

Reviewer: *Table 3, lines 3-4 (legend): Mention which of the rows corresponds to the threshold obtained excluding the Smirnova list.*

Response: The sentence was changed to : "For $\theta_{500hPa} - \theta_{SST}$ **a second** threshold satisfying all polar-low lists beside the one from Smirnova is presented."

Reviewer: *Figure 1 (legend): Modify the first sentence because it is not well written, and it is not clear.*

Response: "Distribution in the parameters used to identify polar lows, these are the polar-low criteria which are successful in distinguishing between all tracks of both hemispheres (black) and polar low from the different lists (colours)." was changed to:

"Parameter distributions for polar lows and mesocyclones from the different lists (colours) and cyclone tracks of both hemispheres (black). These parameters are successful in distinguishing between polar lows and cyclone tracks, hence become the polar-low criteria."

Reviewer: *Figure 2, line 5 (legend): Add "(f)" before "the sea-level".*

Response: Thanks.

Reviewer: *Figure 2 (legend): The figure e needs to be explained in the legend.*

Response: I added: "(e) The lifetime is the duration of the track."

Reviewer: *Figure 7 (legend): "Parameter distributions of polar lows for the different regions" instead of "Parameter distributions polar lows from the different regions".*

Response: Thanks.

Reviewer: *Figure 7 (legend): The figure d needs to be explained in the legend.*

Response: I added: "(d) The lifetime is the duration of the polar low."

Reviewer: *Figure 9 (legend): The quality of the figure needs to be improved.*

Response: Indeed, I included a wrong legend. It was changed to: "Distribution of the different shear categories in the regions marked by boxes in Figure 3."

Reviewer: *Figure 10, line 4 (legend): "are" instead of "is".*

Response: Thanks.

**Response to reviewer 2.**

I thank the referee for the constructive review. The comments are contributing to an overall improved manuscript.

**Specific comments:**

Reviewer: *L. 148-149. For Noer list, which time steps is used track point with hourly resolution or timesteps identified from satellite images? If the author uses hourly data, that might also affect the high match rate.*

Response: L. 133-134 states: "Track points are at hourly resolution due to interpolation between locations identified from satellite images." I do not think that the interpolation has a large influence on the match rate, since the matching requires that the cyclone track is located within 150 km for at least half of the track points of the PL. Interpolation is increasing the amount of track points, however the criteria has to be satisfied for the same fraction of track points (half).

Reviewer: *Table1: I suggest adding region and period for each PL list to this table. This would be helpful for reads.*

Response: Thanks for the suggestion, both is added.

Reviewer: *L. 200. Please add a brief description of how to calculate the vertical shear, especially, vertical levels used for the calculation, although it was given in Stoll et al. (2021)*

Response: A mathematical description of the calculation of the wind shear strength and angle has been added.

Reviewer: *L. 215: PL lists used to derive the PL criteria is only from Northern hemisphere. This might cause some bias in the PLs in southern hemisphere as the environment of the PLs are slightly different between both hemispheres. Although there is not any PL list with sufficient quality for southern hemisphere, I suggest noting that the PL criteria is derived from PLs in only northern hemisphere.*

Response: I formulated the following paragraph to clarify: "The distributions in the parameters utilised for the PL criteria (Fig. 1 are quite similar for the five PL lists from the Northern Hemisphere, expressing that the criteria are independent of the region and the producer of the list. Also, the mesocyclone list of Verezemskaya from the Southern Ocean presented in the supplement qualitatively agrees with the PL lists, pointing towards general applicability of the PL criteria in both hemispheres. Further, the distributions of the PLs and cyclones are distinct from each other, demonstrating that the parameters are successful for separation. "

Reviewer: *L. 215: I suggest that the author explains why these variables were tested for PL criteria before giving the criteria. I suppose that is because these variables are associated with the PL characteristics. The first paragraphs of 3.1.1, 3.1.2, 3.2 ,and 3.3 would suit this purpose.*

Response: I reformulated the beginning of Section 3 for this issue: "Different parameters are compared for their ability to separate between PLs and other cyclones (Tab. 2, Fig. 1, 2). Included are parameters that were found successful by Stoll et al. (2018) and new parameters that are expected to capture the characteristics of PLs or their environments. " I think it is better to keep the motivation of each of the variables at the beginning of the paragraph that describes the derivation of the individual criteria to avoid jumping back and forth in the content.

Reviewer: *Fig.1: Could you indicate the values used for the criteria in the figure?*

Response: The values of the criteria are provided in Table 2 and in the text. Further, the triangle of the list with the weakest threshold indicates the criteria in Figure 1. I think indicating the values in the figure would make it rather more confusing. If you disagree, I would be interested in a specific suggestion on how the values should be indicated.

Reviewer: *L. 245: I do not agree that the U10m > 15m/s can be ignored, because this criterion is important for hazards. However, when this criterion is applied to model outputs, the maximum wind strongly depends on the model grid spacing. Thus, I agree that the authors use vorticity criterion. I suggest removing "and can be ignored when utilizing the relative vorticity"*

Response: The sentence "Hence, $U_{10m} > 15\,\mathrm{m\,s^{-1}}$ may be considered necessary due to the definition of Rasmussen and Turner (2003), but it is not a sufficient criterion for identifying PLs and can be ignored when utilizing the relative vorticity."
is reformulated to: "Hence, a near-surface wind criteria is unnecessary for the detection of PLs when a criteria is applied that ensures a strong mesoscale vortex, as the here utilized relative vorticity criteria."

Reviewer: *L. 387-388: Why both miss and false-positive rates in Smirnova list is higher than those in Golubkin list? Usually there are trade-off relation between miss and false-positive rates.*

Response: I agree that one would expect such a trade-off relation if both lists largely agree on their interpretation of PLs and their method of detection, however disagree on the threshold at which a system is included to the list. However, the interpretation of PLs or method of detection can also be quite different. The Golubkin and Rojo list appear to agree more to each other than to the Smirnova list. For example the former two include synoptic weather maps to their procedure of PL detection, which is not the case for the latter. To clarify this, I added another sentence to the manuscript:

"This expresses that the Rojo and Golubkin list agree more on their interpretation of a PL and detection method then the Rojo and Smirnova list."

Reviewer: *L. 407-409: There are two types of false positives in a reanalysis. One is a cyclone that is detected as a PL but is not considered as reasonable detection. The other is a cyclone that is detected as a PL and considered as reasonable detection but do not exist in the real atmosphere. The author examined only the former. The latter false positive should be evaluated by comparing the detected PLs with observation such as satellite images.*

Response: I agree that it would be beneficial to compare some of the polar tracks to satellite images. Unfortunately the Dundee satellite retrieving station (http://due.esrin.esa.int/page_company69.php) does not operate anymore. I have not found another place where satellite images from high latitudes are provided open source. I would be interested if you could point out one.

Regarding your comment, I think it is unlikely ERA-5 is producing spurious PLs out of nowhere. ERA-5 is fundamentally relying on data assimilation, including satellite images. If a satellite image would display no PL, the data assimilation would "remove" the PL from the ERA-5 product.

ERA-5 is closely connected to the analysis step of the operational forecast from ECMWF. From my observation the operational forecast is not producing spurious non-PLs at the analysis step or the short-term forecast. Hence, I think my evaluation of false-positives is trustworthy even without comparing to satellite imagery.

Reviewer: *L. 444-445: Some references are required for this sentence.*

Response: Indeed, references are added.

**Technical corrections**

Reviewer: *L. 62. effected $\Rightarrow$ affected*
Response: Thanks.
Reviewer: *L. 491: count $\Rightarrow$ are counted*
Response: Thanks.
Reviewer: *L. 620: know $\Rightarrow$ known*
Response: Thanks.
Reviewer: *L. 667: Stratospheric 1 Warmings $\Rightarrow$ Stratospheric Warmings*
Response: Thanks.

---

## Referee Report (RR2)

**Review for the manuscript wcd-2021-60: "A global climatology of polar lows investigated for local differences and wind-shear environments" from Patrick Johannes Stoll**

**Main comment:**

This study focuses on polar lows (PLs) climatology in both Northern and Southern Hemisphere and on finding suitable identification criteria to discard PLs from other mesoscale cyclones. This study makes a significant contribution in referencing and comparing PLs found in the ERA5 reanalysis to previous observational climatologies.

The author corrected the manuscript following the comments of both reviewers, which contributed in improving the manuscript. Although some minor typos remain after the correction, the manuscript is clear and well-structured. Therefore, I would recommend the manuscript for publication after some small minor revision.

**Specific comments:**

- P. 2 L. 54: A quick sentence should be added in the text on why those criteria differ from the ones used in previous studies.
- P. 3. L. 66: Maybe a quick definition of the shear could be added into parenthesis.
- P. 4 L. 92: It should be noted that, although band-pass filtered approaches may include synoptic-scale cyclones, spectral filtering can be used to only retain the wave-numbers equivalent to mesoscale features with similar scales as PLs. This has been successfully done for PLs using spectral filtering to the vorticity field by previous studies such as Zappa et al. (2014), Yanase et al. (2016), Smirnova and Golubkin (2017) and Stoll et al. (2018).
- P. 10 L. 292-293: This sentence is a bit confusing. Maybe rephrase it by stating "criterion" after the "first/second/etc.".
- P. 11 L. 302: A sentence could be added on the fact that these criteria, although relatively efficient, do not detect all observed PLs from the previous lists.
- P. 14 L. 337-340: This is an interesting suggestion: has it been verified for some PLs of this study or in other studies?
- P. 15 L. 378-379: Instead of "punishing" the region, how about decreasing the SST-T500 criterion for the Southern Hemisphere then? Have you tried different thresholds/criteria for SH PLs and if so, would using different thresholds/criteria for NH and SH PLs be more beneficial (since SH PLs seem less "responsive" to your criteria than NH PLs)? Maybe a word on this could be added.
- P. 24 Section 5.2: It would be interesting to know if trends appear in the different PL shear distributions (in the global of each shear distributions and the distributions for each region). If you already calculated it (or if it's not too much work) a sentence or two on this would be appreciated.

**Technical comments:**

- P. 1 L. 18: I think references to books should contain the page numbers corresponding to the citation.
- P. 11 L. 300: I think "Furthermore" might be more appropriate than "Further".
- A few typos (misplaced coma/point/capital letter/e.g.) are found along the text (ex: L. 120, 397).